# Discovery of fungal oligosaccharide-oxidising flavo-enzymes with previously unknown substrates, redox-activity profiles and interplay with LPMOs

Majid Haddad Momeni [1,2], Folmer Fredslund [3], Bastien Bissaro [2], Olanrewaju Raji [4], Thu V. Vuong [4], Sebastian Meier [5], Tine Sofie Nielsen[1], Vincent Lombard[6], Bruno Guigliarelli[7], Frédéric Biaso[7], Mireille Haon[2], Sacha Grisel[2], Bernard Henrissat [6,8,9], Ditte Hededam Welner [3], Emma R. Master[4,10], Jean-Guy Berrin[2✉] & Maher Abou Hachem [1✉]

Oxidative plant cell-wall processing enzymes are of great importance in biology and biotechnology. Yet, our insight into the functional interplay amongst such oxidative enzymes remains limited. Here, a phylogenetic analysis of the auxiliary activity 7 family (AA7), currently harbouring oligosaccharide flavo-oxidases, reveals a striking abundance of AA7-genes in phytopathogenic fungi and Oomycetes. Expression of five fungal enzymes, including three from unexplored clades, expands the AA7-substrate range and unveils a cellooligosaccharide dehydrogenase activity, previously unknown within AA7. Sequence and structural analyses identify unique signatures distinguishing the strict dehydrogenase clade from canonical AA7 oxidases. The discovered dehydrogenase directly is able to transfer electrons to an AA9 lytic polysaccharide monooxygenase (LPMO) and fuel cellulose degradation by LPMOs without exogenous reductants. The expansion of redox-profiles and substrate range highlights the functional diversity within AA7 and sets the stage for harnessing AA7 dehydrogenases to fine-tune LPMO activity in biotechnological conversion of plant feedstocks.

[1] Department of Biotechnology and Biomedicine, Technical University of Denmark, Lyngby, Denmark. [2] INRAE, Aix Marseille Univ, Biodiversité et Biotechnologie Fongiques (BBF), Marseille, France. [3] The Novo Nordisk Center for Biosustainability, Lyngby, Denmark. [4] Department of Chemical Engineering and Applied Chemistry, University of Toronto, Toronto, ON, Canada. [5] Department of Chemistry, Technical University of Denmark, Lyngby, Denmark. [6] Architecture et Fonction des Macromolécules Biologiques, UMR 7257 CNRS, USC 1408, Aix Marseille Univ, Marseille, France. [7] Aix-Marseille Univ, CNRS, UMR7281 Unité de Bioénergétique et Ingénierie des Protéines (BIP), Marseille, France. [8] INRAE, USC1408 Architecture et Fonction des Macromolécules Biologiques (AFMB), Marseille, France. [9] Department of Biological Sciences, King Abdulaziz University, Jeddah, Saudi Arabia. [10] Department of Bioproducts and Biosystems, Aalto University, Espoo, Finland. ✉email: jean-guy.berrin@inrae.fr; maha@bio.dtu.dk

The involvement of oxidative processes in polysaccharide degradation by fungi has been proposed by the pioneering work of Eriksson et al. in 1974[1]. This notion has gained strong support by the recent discovery of lytic polysaccharide monooxygenases (LPMOs) that uniquely catalyse the oxidative cleavage of glycosidic bonds in (semi)crystalline polysaccharides such as starch[2–4], chitin[5], cellulose[6–8] and cellulose-bound hemicelluloses, e.g., xyloglucan[9] and xylan[10]. Besides LPMOs, filamentous fungi co-secrete an impressing diversity of carbohydrate-specific oxidoreductases[11]. To date, only four fungal oligosaccharide-oxidising enzymes from the auxiliary activity family 7 (AA7) in the Carbohydrate Active enZyme (CAZy) database[12], have been characterized. In addition, oligosaccharide oxidases from plants have been reported[13,14], but are currently not assigned into AA7. Our insight is, thus, clearly limited regarding the biological roles as well as the diversity of substrates and redox features within this family. Fungal AA7 enzymes catalyse the oxidation of the reducing end C1-OH in e.g. cellooligosaccharides[15] and lactose[16], xylooligosaccharides[17] as well as chitooligosaccharides[18] to the corresponding lactones. Electrons derived from oligosaccharide oxidation reduce the FAD cofactor that is subsequently re-oxidised via electron transfer to $O_2$ to generate $H_2O_2$ (oxidase activity). Notably, comparable dehydrogenase and oxidase activities have been observed for an AA7 enzyme via electron transfer to artificial electron acceptors instead of $O_2$[16].

The tertiary structures of hitherto described AA7 enzymes[15–17,19] share a common fold comprising an N-terminal FAD-binding domain (F domain) and a C-terminal substrate-binding domain (S domain). This fold is common within the vanillyl alcohol oxidase (VAO, EC. 1.1.3.38) super family[20] that harbours AA7. All previously characterised AA7 oxidases are distinguished by a cysteinyl and histidyl bi-covalently tethered FAD cofactor. By contrast, other VAO family members harbour mostly a mono-covalently (or less commonly a non-covalently) bound FAD cofactor[20].

Reactive oxygen species (ROS), and especially $H_2O_2$, play important roles in lignocellulose degradation by fungi, but the underpinning molecular details of these roles remain poorly understood[11]. Although $O_2$ has long been considered as the co-substrate of LPMOs[5,7], recent findings suggest that $H_2O_2$ is the more favourable co-substrate during polysaccharide oxidative cleavage[21–23]. LPMO catalysis is mediated by a Cu cofactor[6] that must be reduced from Cu(II) to Cu(I) to prime the reaction. Enzymatic priming of LPMOs by the modular pyrroloquinoline-quinone-dependent pyranose dehydrogenase[24] (CAZy family AA12 dehydrogenase domain appended to an AA8 cytochrome $b$ haem domain) and FAD-dependant glucose-methanol-choline (GMC) superfamily of oxidoreductases (AA3), most notably the fungal cellobiose dehydrogenase (CDH) that also possesses a cytochrome $b$ haem domain[25–27], has been reported. The activity on cellooligosaccharides and the transcriptional co-regulation as well as the co-secretion with cellulose active AA9 LPMOs[28], justified extensive studies on CDH as a model for direct electron transfer and priming of LPMOs[25,27,29]. Not all fungi possess the CDH/LPMO pair, suggesting the presence of additional redox partners and mechanisms for LPMO activation. By analogy, the co-secretion of AA7s with LPMOs upon fungal growth on plant biomass[30,31] prompted us to hypothesize a redox interplay between these two enzyme classes.

Here, we report phylogenetic analyses, combined with the selection and characterisation of five fungal AA7s, three of which belong to previously unexplored clades. We demonstrate activity on four saccharides, previously not reported as AA7 substrates. In-depth analysis of a cellooligosaccharide dehydrogenase with a mono histidyl-tethered FAD highlights clade-dependant redox-profiles

within AA7. Importantly, we unveil direct activation and potentiation of LPMO activity on cellulose by this newly-discovered dehydrogenase. This study provides biochemical, structural and mechanistic insights into AA7 enzymes as components of the fungal redox network secreted during growth on biomass. Our findings suggest a way to tune LPMO activity for enzymatic degradation of major recalcitrant polysaccharides using AA7 dehydrogenases.

## Results

**Clade-dependant variations in the FAD covalent tethering residues in AA7.** To date, the molecular specificity signatures in AA7s have not been unveiled. To explore the sequence diversity in this family, we used the sequence of the previously characterised *Fusarium graminearum* chitooligosaccharide oxidase *Fg*ChitO[18] as a query to retrieve sequences comprising 470–570 amino acids (aa) from a BLAST search. The sequences ($n = 1927$), originating from fungi, eukaryotic microorganisms and plants, were aligned and curated to generate a phylogenetic tree formed by six clades (Fig. 1a). Clade I, which is the largest (34% of all sequences), is dominated by plant and fungal sequences (Fig. 1b). Indeed, this clade contains plant non-carbohydrate active enzymes, e.g., the berberine bridge enzyme from *Eschscholzia californica Ec*BBE[32] and the monolignol oxidase from *Arabidopsis thaliana At*BBE-like 15[33], as well as the oligogalacturonide oxidase from *A. thaliana At*OGOX1[13], although none of these sequences are currently assigned into AA7.

The sequences in clades IIa, III and V are mainly from Ascomycota, whereas mostly Basidiomycota sequences populate clades IV and VI. Remarkably, the majority of the retrieved Ascomycota sequences were from genera known to harbour phytopathogens, e.g., *Fusarium*, *Magnaporthe*, *Colletotrichum*, *Bipolaris*, *Alternaria* or *Botrytis*. The enrichment of phytopathogen sequences is striking in clade II. Thus, fungus-like eukaryotic plant pathogens from Oomycota clustered in clade IIb (90.5% of sequences), whereas the remaining sequences in clade II originate mainly from Ascomycota phytopathogens. All the four previously described AA7 fungal oligosaccharide oxidases clustered in a branch of clade V (Va) (Fig. 1a). The scarce insight into AA7 is evident from the lack of characterized members from (sub)clades II, III, IV, Vb and VI.

Since bi-covalent FAD tethering has been the hallmark of all hitherto characterised AA7s, we analysed the conservation of the cysteine and histidine FAD-tethering residues across the phylogenetic tree. Strikingly, clade II harboured exclusively non-canonical sequences with substitution of the FAD-binding cysteine, histidine or both, while these residues were highly conserved in the other clades (Fig. 1c). We selected five AA7 candidates belonging to different clades (Fig. 1a, Supplementary Table 1) for recombinant expression and functional characterization.

**Identification of a strict oligosaccharide dehydrogenase in AA7.** The selected fungal sequences share 25–39% amino acid sequence identity and originate from *Aspergillus nidulans* (*An*AA7A, clade I), *Fusarium graminearum* (*Fg*CelDH7C, clade IIa and *Fg*Chi7B, clade Va), *Magnaporthe oryzae* (*Mo*Chi7A, clade Vb) and *Polyporus brumalis* (*Pb*Chi7A, clade VI) (Supplementary Table 1). The selected AA7 enzymes were successfully expressed in *Pichia pastoris* and highly pure enzymes were obtained from a single affinity purification. The substrate specificity of each recombinant enzyme was assessed against a panel of 40 compounds including saccharides with a degree of polymerization (DP) 1–4 as well as sugar alcohols and aromatic alcohols (Supplementary Table 2) by monitoring both their oxidase activity (i.e. $H_2O_2$ production using a peroxidase coupled assay) and dehydrogenase

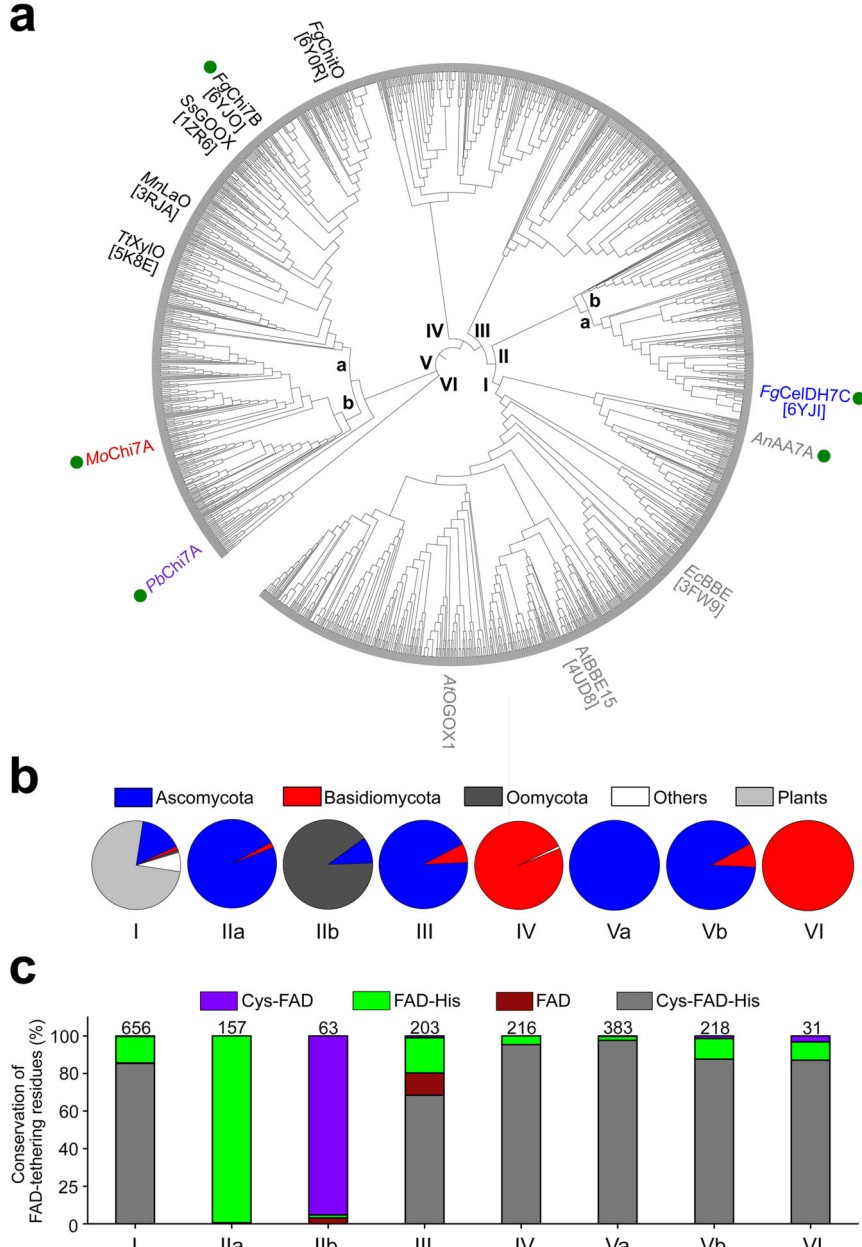

**Fig. 1 Phylogenetic analysis of AA7-like sequences. a** The phylogenetic tree is based on 1927 sequences. Biochemically characterized enzymes are coloured according to clade, with green circles indicating enzymes from the present study. The PDB entries (in square brackets) are given for available enzyme structures. Clade Va harbours the canonical previously described oligosaccharide oxidases: SsGOOX[15] from *Sarocladium strictum* active on cellooligosaccharides, MnLaO[16] from *Microdochium nivale* active on lactose, TtXylO[17] from *Thermothelomyces thermophilus* active on xylooligosaccharides and FgChitO[18] from *Fusarium graminearum* active on chitooligosaccharides. Clade I contains characterised plant enzymes[33,39] that are not assigned in AA7 including the oligogalacturonide oxidase (AtOGOX1)[13] from *Arabidopsis thaliana*[13,33,39]. **b** Clade-wise taxonomic distribution of putative AA7 sequences. **c** Clade-wise conservation percentage of the histidine and cysteine FAD-cofactor tethering residues. The number of sequences within each clade is indicated above each bar. The accessions of the sequences in the tree are provided in the Source Data file.

activity using 2,6-dichlorophenolindophenol (DCIP) as an artificial electron acceptor.

FgChi7B (clade Va), MoChi7A (clade Vb) and PbChi7A (clade VI) displayed the highest normalised rates ($V_o$/E) on chitooligosaccharides, followed by N-acetyl glucosamine (GlcNAc) (Supplementary Fig. 1). Importantly, we also discovered oxidase activity against substrates not reported before in AA7. Thus, MoChi7A was active on N-acetyl galactosamine (GalNAc), lacto-N-biose (LNB) and galacto-N-biose (GNB), while PbChi7A oxidised mannooligosaccharides (Supplementary Table 3, Supplementary Fig. 1). The $K_M$ values of MoChi7A and PbChi7A

towards GlcNAc were about 20-fold lower compared to FgChi7B (Table 1), highlighting a marked difference in affinity for monosaccharides between these enzymes. No activity was detected for AnAA7A on any of the tested substrates, suggesting that AA7 targets a wider range of substrates than currently reported. The pH-dependence of activity profiles showed the highest relative activities in the pH range 7.0–7.5, except for PbChi7A, which appeared to have a broader profile (Supplementary Fig. 2).

Apart from FgCelDH7C (*vide infra*), the dehydrogenase activity for a given substrate was in the same range or higher

**Table 1 Apparent kinetic parameters of AA7-catalysed oxidation of saccharides.**

**Oxidase kinetics**

| Substrate | Parameter | Enzyme | | |
|---|---|---|---|---|
| | | **MoChi7A** | **PbChi7A** | **FgChi7B** |
| GalNAc | $k_{cat}$ (s$^{-1}$) | 3.7 ± 0.1 | ND | ND |
| | $K_M$ (mM) | 1.69 ± 0.12 | ND | ND |
| | $k_{cat}/K_M$ (s$^{-1}$ M$^{-1}$) | 2.16 × 10$^3$ | | |
| GlcNAc | $k_{cat}$ (s$^{-1}$) | 3.8 ± 0.1 | 3.0 ± 0.2<br>3.3 ± 0.2[a] | 6.8 ± 0.3 |
| | $K_M$ (mM) | 0.80 ± 0.05 | 0.69 ± 0.20<br>0.50 ± 0.10* | 15.7 ± 1.13 |
| | $k_{cat}/K_M$ (s$^{-1}$ M$^{-1}$) | 4.75 × 10$^3$ | 4.27 × 10$^3$<br>5.94 × 10$^{3a}$ | 4.30 × 10$^2$ |
| Chitobiose | $k_{cat}$ (s$^{-1}$) | 4.4 ± 0.1<br>18.3 ± 0.4[a] | 3.8 ± 0.04 | 2.3 ± 0.03<br>28.7 ± 0.4 |
| | $K_M$ (mM) | 0.064 ± 0.01<br>0.33 ± 0.03[a] | 0.3 ± 0.02 | 0.27 ± 0.01<br>1.32 ± 0.07 |
| | $k_{cat}/K_M$ (s$^{-1}$ M$^{-1}$) | 6.83 × 10$^4$<br>5.56 × 10$^{4a}$ | 1.27 × 10$^4$ | 8.58 × 10$^3$<br>2.17 × 10$^{4a}$ |
| Chitotriose | $k_{cat}$ (s$^{-1}$) | 3.71 ± 0.09 | 3.55 ± 0.04<br>2.0 ± 0.1[a] | 6.14 ± 0.29 |
| | $K_M$ (mM) | 0.09 ± 0.01 | 0.28 ± 0.01<br>0.75 ± 0.1[a] | 0.25 ± 0.03 |
| | $k_{cat}/K_M$ (s$^{-1}$ M$^{-1}$) | 4.22 × 10$^4$ | 1.25 × 10$^4$<br>2.67 × 10$^{3a}$ | 2.44 × 10$^4$ |

**Dehydrogenase kinetics of FgCelDH7C.**

| | **Cellobiose** | **Cellotriose** | **Cellotetraose** | **Cellopentaose** |
|---|---|---|---|---|
| $k_{cat}$ (s$^{-1}$) | 30.8 ± 0.81 | 36.1 ± 0.74 | 39.27 ± 0.76 | 45.28 ± 2.03 |
| $K_M$ (mM) | 5.35 ± 0.39 | 3.94 ± 0.22 | 6.06 ± 0.27 | 11.25 ± 0.94 |
| $k_{cat}/K_M$ (s$^{-1}$ M$^{-1}$) | 5757 | 9162 | 6480 | 4025 |

[a]Apparent kinetic parameters using the dehydrogenase assay. The kinetic parameters are determined from a global fit of the Michaelis-Menten expression to triplicate ($n = 3$ independent experiments), except for cellopentaose ($n = 1$ experiment). The data are shown as means ± standard deviations. For cellopentaose the error estimates of the fit to the single data set are shown.

(up to 12-fold) than the corresponding oxidase activity, and the substrate specificity profiles were mostly similar using both assays (Table 1, Supplementary Fig. 1). By contrast, FgCelDH7C acted as a dehydrogenase with a preference for cellooligosaccharides with $k_{cat}$ values of 32–42 s$^{-1}$ and the highest catalytic efficiency on cellotriose (Table 1, Supplementary Fig. 1h). The enzyme was also active on glucose, α-(1,4)-glucooligomers (maltooligosaccharide) and lactose, all sharing a reducing end glucosyl unit. Interestingly, the oxidase activity was estimated to be several orders of magnitude lower ($V_o/E \approx 1 - 10 \times 10^{-5}$ s$^{-1}$) (Supplementary Fig. 1g, h). This uniquely higher dehydrogenase/oxidase activity ratio compared to canonical AA7s distinguishes FgCelDH7C as an AA7 dehydrogenase with only trace oxidase activity. Interestingly, FgCelDH7C had a markedly different flavin absorbance spectrum than typical oxidase homologues (Fig. 2a), suggestive of large changes in the FAD chemical environment. The stability of the enzyme was also evaluated to verify the structural integrity (unfolding temperature $T_m > 55$ °C, Supplementary Fig. 2b). Both MoChi7A, which catalyses efficient oxidation of substrates not reported within AA7, and the strict dehydrogenase FgCelDH7C were selected for further in situ NMR spectroscopy to bring insight into their redox chemistry. Initial experiments identified the chemical shift assignments of substrates, intermediates and products based on two-dimensional $^1$H-$^{13}$C NMR analyses (Supplementary Fig. 3). Time-resolved two-dimensional $^1$H-$^{13}$C HSQC NMR spectra were acquired to follow the time-course conversion of GlcNAc and GalNAc by MoChi7A, showing the initial formation of a 1,5-pyrano (δ) lactone prior to rearrangement to a 1,4-furano (γ) lactone. The carboxylic acid open form was the main product of GlcNAc

oxidation (due to lactone hydrolysis), whereas the 1,4-furano (γ) lactone accumulated as the main product of GalNAc oxidation (Supplementary Fig. 4). Similarly, assignments were established for the oxidation of cellobiose by FgCelDH7C (Supplementary Fig. 5). A time series of one-dimensional $^1$H NMR spectra was used for kinetic analysis of the FgCelDH7C catalysed oxidation of cellobiose in the presence of DCIP that displayed exchange line broadening during the reaction (Supplementary Fig. 6a). Line shape analysis showed that oxidised DCIP was reduced in a slow to intermediate exchange regime (exchange rate ≈ 87 s$^{-1}$) (Supplementary Fig. 6b). Interestingly, the sharp signals of reduced DCIP were only observed upon stoichiometric conversion of the oxidized DCIP form and hydrolysis of the lactone to the acid (Supplementary Fig. 6a), most likely due to the irreversible hydrolysis breaking the equilibrium. Notably, the oxidase reaction with O$_2$ as an electron acceptor occurred orders of magnitude slower in the absence of DCIP as electron acceptor (Fig. 2b, c), corroborating the activity data from the coupled peroxidase assay above. Further NMR analyses and activity assays validated the dependence of cellooligosaccharide oxidation on DCIP concentration (Supplementary Fig. 7).

**Structural elements underpinning the dehydrogenase/oxidase activity profiles.** We determined the crystal structures of FgChi7B and FgCelDH7C at resolutions of 2.4 and 1.6 Å, respectively (Supplementary Table 4, Supplementary Fig. 8). Both enzymes share the canonical AA7 fold comprising an FAD-binding F domain and a substrate-binding S domain that is formed by a central β-sheet flanked by α-helices (Supplementary Fig. 9a, b). Moreover, the two enzymes share an aromatic cluster comprising: (i) the catalytic base tyrosine (FgCelDH7C Y454), (ii) a tyrosine/phenylalanine (FgCelDH7C F99) that stacks onto and stabilises the catalytic base, and (iii) an aromatic residue that stacks mainly onto the saccharide unit penultimate to the reducing end (FgCelDH7C F383) (Fig. 3a). Interestingly, the residues of this aromatic cluster are also highly conserved in most sequences in clades IIa and Va (Fig. 3b), highlighting their importance in catalysis and substrate binding.

By contrast, the structure of FgCelDH7C revealed a unique active site architecture, as compared to all hitherto available AA7 structures. The first notable overall structural difference is that a loop (residues Q335-F346) that flanks the active site in FgCelDH7C appears shorter due to a preceding helical segment that folds away from the active site, which provides a less occluded and more solvent-exposed FAD-cofactor compared to FgChi7B and all hitherto reported AA7 structures (Supplementary Fig. 9).

Strikingly, the mode of FAD-anchoring represents a second unique characteristic distinguishing FgCelDH7C from the other AA7 structures. Thus, the isoalloxazine ring of the FAD cofactor in FgChi7B is bi-covalently anchored at the C6 atom to C162 S$^\gamma$ (6-S-cysteinyl) and the 8α-methyl group to H102 N$^{\delta 1}$ (8α-N1-histidyl), akin to all previously determined AA7 structures (Supplementary Fig. 9a). By contrast, FgCelDH7C is mono-covalently histidyl-tethered due to the substitution of the FAD-tethering cysteine to a glycine (Fig. 3c). Notably, this cysteine is conserved in the majority of AA7 except in clade II sequences that possess a glycine, an alanine or a serine at this position (Fig. 3b, d, Supplementary Fig. 10). A third structural signature of FgCelDH7C, compared to all available AA7 structures, is the substitution of a histidine residue facing the isoalloxazine ring in the FAD-binding domain to a serine (S165) (Fig. 3c).

A fourth different motif in FgCelDH7C entails four arginine residues that protrude into the substrate-binding pocket (Fig. 3c, d), providing a distinctively positive charged milieu compared to other structurally characterised AA7s (Supplementary Fig. 11).

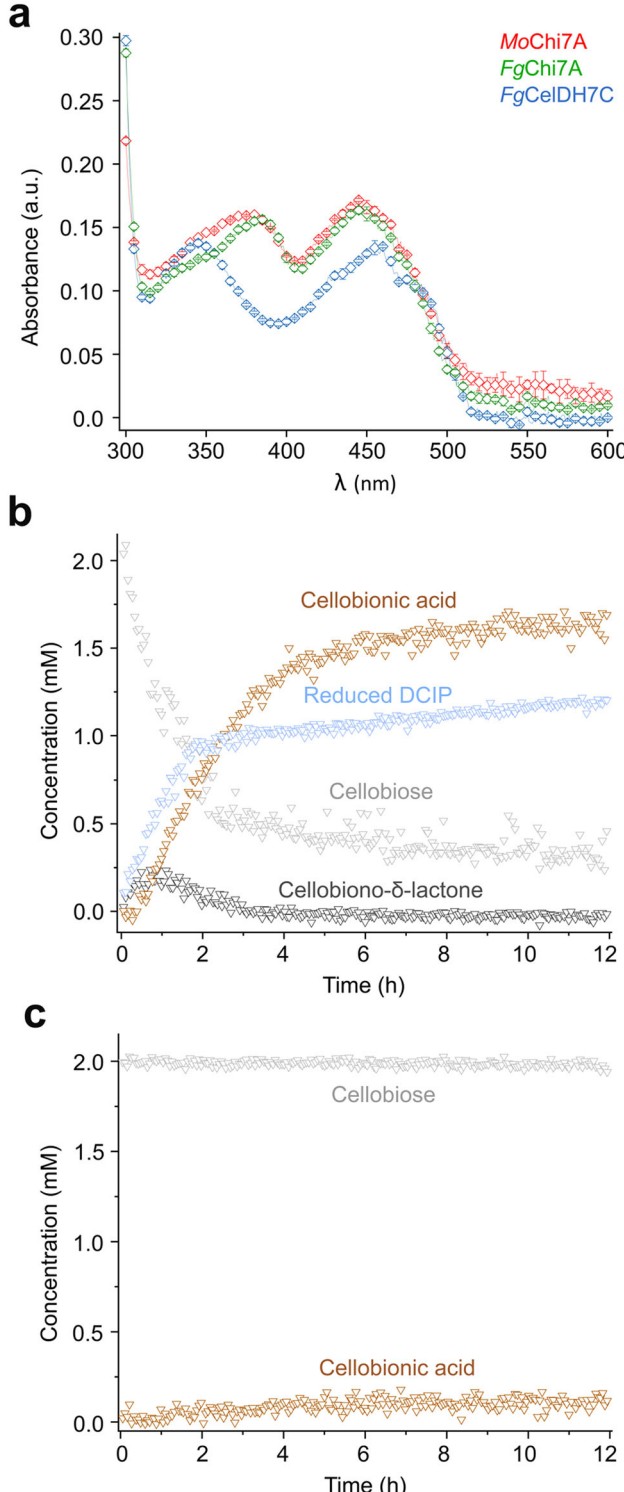

**Fig. 2 Spectral properties and time-course NMR analysis of the AA7 cellooligosaccharide dehydrogenase *Fg*CelDH7C. a** Spectral comparison of the flavin absorbance of *Fg*CelDH7C and the oxidases *Fg*Chi7B as well as *Mo*Chi7A, all at 20 μM. The data are means ± standard deviations (*n* = 3 independent experiments). **b** and **c** Show the time resolved in situ $^1$H NMR analyses of the oxidation of cellobiose by *Fg*CelDH7C in the presence (**b**) or absence (**c**) of 1.3 mM 2,6-dichlorophenolindophenol (DCIP) as an electron acceptor, respectively. The presented spectra in panels b and c are from a single experiment (*n* = 1). Source data are provided as a Source Data file.

Notably, members of clade IIb share the above signatures with clade IIa, except for the inverted substitution of the FAD-tethering residues. Hence, the cysteine is conserved, but the histidine counterpart is substituted for an arginine (Fig. 1, Supplementary Fig. 10). In conclusion, the present analysis allows the identification of active-site signatures associated with the functional variation in AA7, e.g., activity on oligosaccharides and the dehydrogenase versus the typical oxidase activity.

**The cellooligosaccharide dehydrogenase *Fg*CelDH7C primes and fuels LPMO activity**. To investigate if AA7 enzymes can trigger the activity of LPMOs, we set up AA7-LPMO mediated cellulose degradation assays to monitor the release of native (unmodified) and oxidised cellooligosaccharides from cellulose. The two cellulose-active LPMOs from *Podospora anserina* *Pa*LPMO9E and *Pa*LPMO9H[34] were used in these assays. The interplay between *Fg*CelDH7C and the C1-oxidising *Pa*LPMO9E was evident from the marked increase in native and C1-oxidised cellooligosaccharides released from phosphoric acid swollen cellulose (PASC) relative to the control reaction, which was fuelled by electrons from ascorbate (Supplementary Fig. 12a). Since *Fg*CelDH7C is inactive on cellulose, the release of oligosaccharides in this assay is solely attributed to the *Pa*LPMO9E activity. However, the generation of C1-oxidised cellooligosaccharides, by both *Fg*CelDH7C and *Pa*LPMO9E, precluded determining the contribution of each enzyme to the total amount of oxidized species. Therefore, a similar assay was also performed with the C4-oxidising *Pa*LPMO9H (instead of *Pa*LPMO9E) as it allowed attributing both C4 and C1/C4 oxidised cellooligosaccharides (identification based on the previous work[34]) exclusively to the LPMO activity. This assay with *Pa*LPMO9H resulted in a significant increase in single- and double-oxidized oligosaccharides relative to the ascorbate control (Supplementary Fig. 12b). We performed similar assays on Avicel (higher crystallinity cellulose than PASC), which revealed that the addition of the preferred *Fg*CelDH7C substrate cellotetraose (DP4) results in a considerable increase in the level of released cellooligosaccharides in a dose-dependent manner (Supplementary Fig. 13). A similar fuelling of LPMO activity was also observed when the same assay was performed using the previously characterised C4-oxidising AA9 LPMO from *Lentinus similis* (*Ls*AA9A)[35,36] (Supplementary Fig. 14). These findings indicate that the ability of *Fg*CelDH7C to fuel cellulose active LPMOs is not LPMO-specific.

We further used the CDH from *P. anserina* to benchmark the *Fg*CelDH7C-*Pa*LPMO9H system, as CDHs are recognised as key redox partners to AA9 LPMOs[26,27,34]. The observed release of cello-oligomers was markedly higher when the LPMO reaction was fuelled by a 3-fold lower *Fg*CelDH7C concentration compared to *Pa*CDHB (Fig. 4, Supplementary Fig. 15). Indeed, the *Fg*CelDH7C-*Pa*LPMO9H pair released 1.5- and 2-fold higher C1-C4 double-oxidised and native cellooligosaccharides, respectively, as compared to the *Pa*CDHB-fuelled reaction. To evaluate if this interplay was specific for *Fg*CelDH7C, we used the promiscuous oxidase *Mo*Chi7A in a similar cellulose degradation assay. Interestingly, the high level of AA7-fuelled LPMO activity appeared specific to *Fg*CelDH7C based on the low amounts of C1- and C1-C4 oxidised species observed in assays where *Fg*CelDH7C was replaced with *Mo*Chi7A that displays low cellooligosaccharides oxidase side-activity (Supplementary Fig. 16).

**Mechanistic insights into the LPMO-AA7 interplay**. To investigate the mechanism of the AA7-LPMO interplay, we analysed the effect of $H_2O_2$ or the superoxide species ($O_2^{\bullet-}$) by performing the reaction in the presence of either horseradish peroxidase (HRP) that converts $H_2O_2$ to $H_2O$[21] or superoxide dismutase

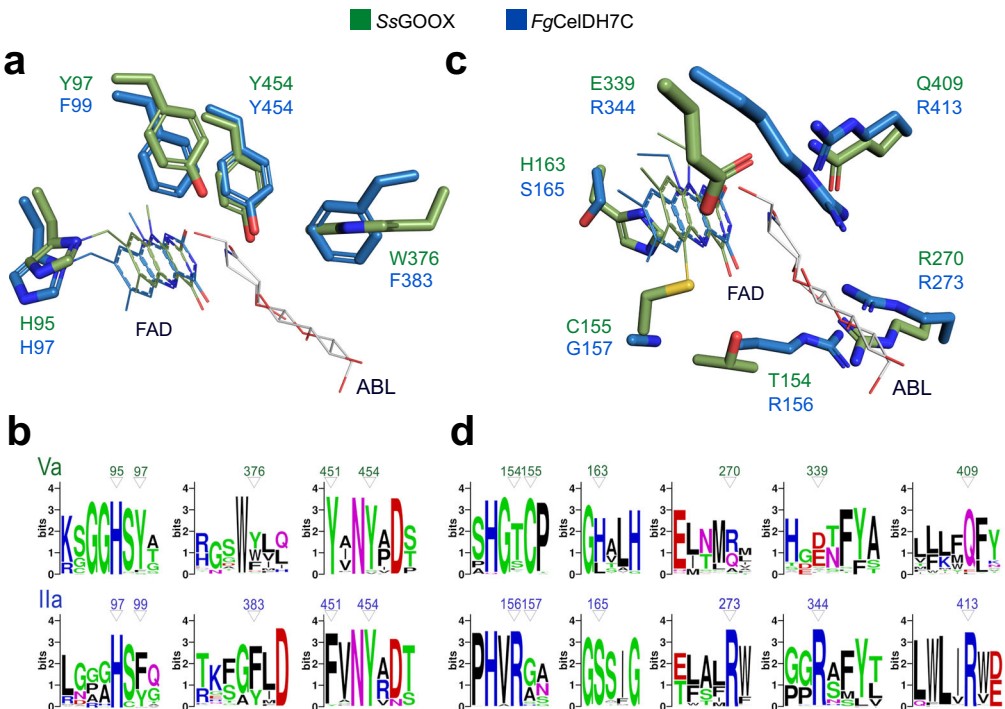

**Fig. 3 Active site signatures of AA7 oligosaccharide oxidases versus dehydrogenases.** The active sites of the clade IIa discovered dehydrogenase *Fg*CelDH7C (PDB: 6YJI) and the canonical clade Va cellooligosaccharide oxidase *Ss*GOOX (in complex with 5-amino-5-deoxy-cellobiono-1,5-lactam ABL, PDB: 2AXR) are shown. **a** The FAD-tethering histidine in addition to an aromatic cluster comprising the tyrosine base catalyst, a phenylalanine/tyrosine and the substrate-stacking aromatic residue are conserved features in fungal oligosaccharide dehydrogenases and oxidases. **b** Sequence logos of patches spanning the structurally similar active site residues shown in (**a**). **c** Active site differences between clades IIa and Va. **d** Sequence logos of patches spanning the structurally divergent active site residues shown in c from clades IIa and Va. The amino acid numbering of the deposited protein sequences is used in the figure.

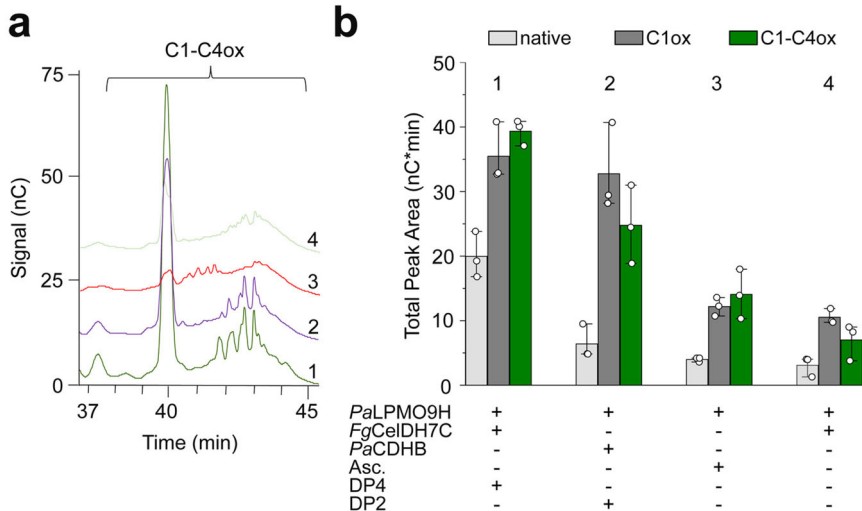

**Fig. 4 Analysis of AA7-LPMO interplay in cellulose degradation.** Reactions were performed on Avicel using *Fg*CelDH7C and *Pa*LPMO9H with subsequent ionic chromatography (HPAEC-PAD) analysis. **a** A representative chromatogram part showing C1-C4 double-oxidised species in the cellulose degradation assay including combinations of Avicel (5 mg mL$^{-1}$), *Fg*CelDH7C (0.4 µM), *Pa*LPMO9H (4 µM), *Pa*CDHB (1.2 µM), cellotetraose (DP4, 0.8 mM) and ascorbate (Asc., 1 mM) as indicated in the figure. **b** Comparison of the cellulose degradation assay based on the cumulative area under the peaks of native (DP3, DP5 and DP6), C1 oxidised (except DP2 and DP4 which were added as substrates for the CDH and AA7, respectively) and C1-C4 double-oxidised cellooligosaccharides from the reactions in (**a**). The data in (**a**) and (**b**) (*n* = 3 independent reactions) were generated in NaOAc/NaOH buffer (50 mM, pH 5.2) at 35 °C. The bar plot in (**b**) shows the means of total peak area (*n* = 3 independent reactions, each shown as a white circle) with standard deviations. Source data are provided as a Source Data file.

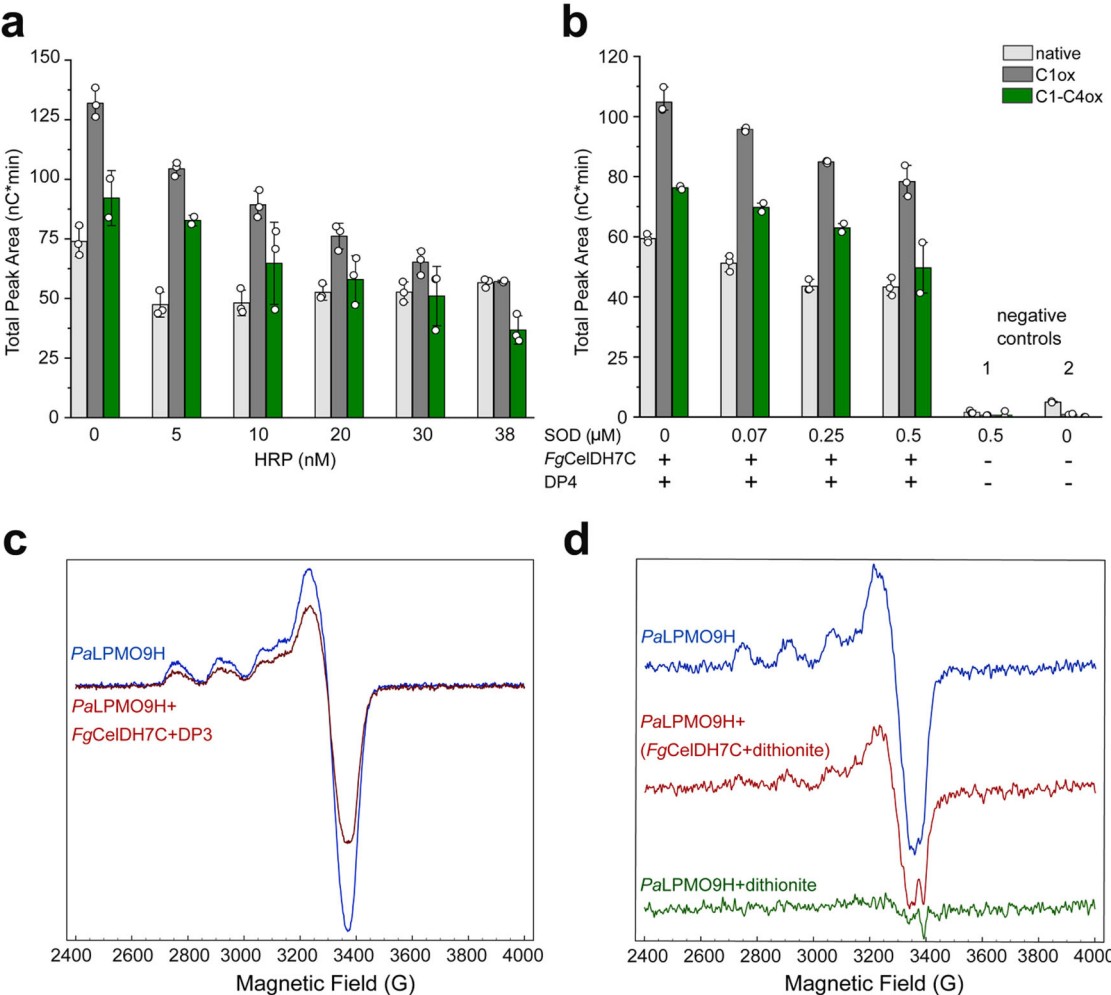

**Fig. 5 Mechanistic insights into the activation of LPMO by *Fg*CelDH7C. a, b** Effect of horseradish peroxidase (HRP) and superoxide dismutase (SOD) on the interplay between *Pa*LPMO9H (4 μM) and *Fg*CelDH7C (0.41 μM) in Avicel degradation based on the total area of native, C1 oxidised (C1 ox) and double C1-C4 oxidised (C1-C4ox) cello-oligosaccharides as analysed by HPAEC-PAD. Controls prepared in the absence of *Fg*CelDH7C (1) or in the absence of both SOD and *Fg*CelDH7C (2). The assays were performed for 18 h at 35 °C and terminated using NaOH (0.1 M) prior to the HPAEC-PAD analyses. The total peak area (white circles, $n = 3$ independent experiments) are shown and the bar plots display the means with standard deviations. **c** X-band Electron Paramagnetic Resonance (EPR) spectra of *Pa*LPMO9H−Cu(II) (100 μM) in the presence of cellotriose (DP3, 1 mM) before (blue line) and after (red line) addition of *Fg*CelDH7C (AA7). **d** X-band EPR spectra of *Pa*LPMO9H−Cu(II) (20 μM) in buffer (blue line), in the presence of *Fg*CelDH7C (AA7) pre-reduced with dithionite (red line), or directly fully reduced with dithionite (10 eq., green line). All EPR solutions and experiments were performed under anaerobic conditions. Samples were in 50 mM NaOAc, pH 5.2 and spectra were recorded at 50 K with a 4 mW microwave power and a 30 Gauss modulation amplitude. The data in (**c**) and (**d**) are based on a single experiment ($n = 1$). Source data for the a and b panels are provided as a Source Data file.

(SOD) that converts superoxide ions[37] to $H_2O_2$ and $O_2$, respectively. The addition of increasing concentrations (5–38 nM) of HRP, reduced the AA7-potentiated LPMO activity (Fig. 5a). These results are consistent with $H_2O_2$, which is supplied by *Fg*CelDH7C and by the side-activity of primed LPMOs, being a preferred LPMO co-substrate. In the presence of increasing concentrations (0.07–0.5 μM) of SOD[37], we observed a similar inhibitory trend (Fig. 5b). These findings suggest that $O_2^{•−}$ plays a specific role in the *Fg*CelDH7C-fuelled LPMO reaction (Fig. 5b). Indeed, the priming reduction of LPMOs with the superoxide ion has been previously proposed[38]. In addition to $O_2^{•−}$-mediated priming, we also investigated the possibility of direct priming electron transfer between the AA7 and the LPMO using Electron Paramagnetic Resonance (EPR) spectroscopy. The extent of LPMO-Cu(II) reduction to Cu(I) was monitored as a decrease of the Cu(II) EPR signal due to the silence of LPMO-Cu(I) species in EPR[25]. The addition of *Fg*CelDH7C to *Pa*LPMO9H pre-incubated with

cellotriose (DP3) under anaerobic conditions led to a decrease (35–40%) in *Pa*LPMO9H-Cu(II) signal (Fig. 5c), whereas DP3 alone failed to induce a change in the Cu(II) EPR signal intensity, consistent with the involvement of *Fg*CelDH7C in the observed priming of *Pa*LPMO9H (Fig. 5d). The fact that only partial reduction of the Cu(II) center was observed could be due to kinetic or thermodynamic barriers. We have previously determined the redox potentials of the LPMOs *Pa*AA9E and *Pa*AA9H at pH 5 to +155 mV and +326 mV, respectivley[25]. The redox potentials of canonical AA7 oxidases with a bi-covalently anchored-FAD is about +130 mV, whereas mutants that abolish the cysteinyl-FAD bond exhibited redox potentials in the 60 mV range[39,40]. Accordingly, the reduction of LPMOs by *Fg*CelDH7C is likely to be thermodynamically feasible assuming a redox potential in the same range as the abolished cysteinyl-FAD bond mutants. To test the possibility of a direct electron transfer between *Fg*CelDH7C and *Pa*LPMO9H, we used a strategy

previously developed to analyse electron transfer between redox partner proteins[41]. We first established that *Pa*LPMO9H-Cu(II) could be efficiently reduced by the chemical reductant dithionite (Fig. 5d). Then, equimolar solutions of *Fg*CelDH7C and *Pa*LPMO9H were prepared anaerobically. The *Fg*CelDH7C solution was pre-reduced with sub-stoichiometric (80%) amounts of dithionite to avoid excess dithionite in the medium. Identical volumes of the two protein solutions were then mixed which resulted in about 50% reduction of the LPMO active site from Cu (II) to the Cu(I) state (Fig. 5d). Altogether, our data are indicative of a molecular oxygen-independent, direct electron transfer from the AA7-bound FAD to the active site of the LPMO.

## Discussion

This study sheds light on an enigmatic family of flavo-enzymes mainly occurring in fungi and fungi-like eukaryotic micro-organisms as well as in plants. Our phylogenetic analysis mapped the four hitherto described AA7 fungal oligosaccharide oxidases together with a large set of mostly uncharacterised orthologues into six clades. The majority of clade I sequences share a highly conserved motif involving the catalytic tyrosine base and a stacking tyrosine (or phenylalanine) that is likely to rigidify the position of the catalytic base (Fig. 3a, Supplementary Fig. 10). Curiously, a few atypical enzymes lack the catalytic tyrosine, e.g., *Ec*BBE from clade I possess a histidine and an adjacent glutamic acid at the equivalent position in the structure. This is in agreement with an alternative mechanism, involving acid/base catalysis previously proposed for *Ec*BBE[42]. Nonetheless, the conservation of the catalytic motif in most sequences supports the conservation of the canonical mechanism across all AA7 clades. These findings expand significantly the sequence inventory that can be considered as AA7.

The substrate aromatic-stacking platform (Phe, Tyr or Trp) (Fig. 3a, Supplementary Fig. 10), is conserved in all previously and presently described oligosaccharides-active AA7s, in addition to a clade I enzyme that is specific for aromatic alcohols[33], indicative of shared substrate-stacking function of the above aromatic platform. By contrast, sequences that lack the aromatic-stacking residue are unlikely to be active on oligosaccharides, e.g., *Ec*BBE, which features in alkaloid biosynthesis[32]. In conclusion, the above described aromatic-stacking platform offers a signature for the identification of putative carbohydrate-active AA7 members, which are likely to have more polar substrate-binding pockets than counterparts active on aromatic substrates. An active site arginine (*Fg*CelDH7C R273) (Fig. 3c, d) has been previously proposed as a specificity signature of carbohydrate-active AA7 members[18]. This arginine correlates with activity on substrates that lack an *N*-acetyl C2-substituent, e.g., cello-oligosaccharides and xylo-oligosaccharides. A glutamine or smaller amino acids (Fig. 3d) at this position allow the accommodation of *N*-acetylated substrates, e.g., chito-oligosaccharides consistent with our structural analysis of *Fg*Chi7B and the previously characterised *Fg*ChitO[18,19] (Supplementary Fig. 17).

Interestingly, *Fg*CelDH7C offers structural and functional insight into the previously unknown clade II (Fig. 3b, d), which is exclusively populated with atypical sequences of enzymes with potentially mono-histidyl (clade IIa), mono-cysteinyl or non-covalently bound FAD (clade IIb). The lower (<10,000-fold) oxidase activity of *Fg*CelDH7C compared to canonical AA7s correlated with conspicuous changes in its active site architecture. The loss of cysteinyl-FAD is likely to elicit a large decrease in the midpoint redox potential, as observed in *Ec*BBE[39] (clade I) and *Ss*GOOX[40] (clade Va) when the cysteinyl bond is abolished[39]. In both cases, the redox potentials decreased from about 130 mV to 53–61 mV accompanied by about 20-fold decrease in oxidase

activity. The changes in the flavin absorbance spectrum due to the loss of the cysteinyl-bond in both enzymes were similar to the *Fg*CelDH7C spectrum (Fig. 2a), which provides a possible spectroscopic feature for lower redox-potential AA7 enzymes.

Oxygen gating and activation at the *re*-side of the isoalloxazine ring in the FAD-domain are also important for the oxidase activity. The entry of $O_2$ to the active site in flavo-enzymes is favoured through hydrophobic tunnels rather than the solvent-filled substrate-binding pocket[43]. These tunnels converge to a cavity sterically gated by apolar residues[33,44], where the $O_2$ is positioned and activated for hydride transfer. Strikingly, we unveiled that the milieu of the FAD, particularly on the *re*-side, is considerably different in clade II as compared to canonical AA7s. An eye-catching substitution of a histidine, which is highly conserved in all other clades, to a serine, leucine or valine in clade II is observed (Fig. 6a, b, Supplementary Fig. 10, sequence patch 3). Mutation of a histidine at the equivalent position in the FAD-dependant aryl alcohol oxidase from *Pleurotus eryngii* was shown to reduce the rate of the oxidative half-reaction by an order of magnitude[45]. The histidine was proposed to contribute to the positioning and polarisation of $O_2$[45], thereby reducing its activation-free energy barrier. The substitution of this histidine in clade II enzymes is accompanied with other marked changes in the $O_2$ binding cavity (Fig. 6a, b). These steric and chemical changes are likely to increase the energy barrier for hydride transfer to $O_2$[43,44], thereby severely reducing the oxidase activity. On the other hand, the uniquely open and positively charged active site of *Fg*CelDH7C, compared to structural homologues, may favour the accommodation of organic electron acceptors to promote the dehydrogenase oxidative half reaction. Taken together, these findings give insights into the structural elements associated with the distinctively high dehydrogenase/oxidase activity profile of *Fg*CelDH7C and possibly AA7 homologues in clade II.

Clade I plant enzymes catalyse diverse reactions, e.g., in alkaloid synthesis[32] and lignin building block synthesis demonstrated for the monolignol oxidase from *Arabidopsis*[33]. By contrast, the biological roles of oligosaccharide-specific oxidases have remained an enigma. Both canonical AA7 oligosaccharide-specific oxidases and clade II dehydrogenases are ubiquitous in phytopathogenic fungi and Oomycota plant parasites, responsible for some of the most devastating plant diseases in agriculture. Recently, the activity of oligogalacturonide oxidases (clade I) from *Arabidopsis* has been reported to reduce the elicitor activity of oligogalacturonides that trigger plant immune-response[13]. Interestingly, plants overexpressing the oligogalacturonide oxidase *At*OGOX1 were more resistant to infection by the phytopathogen *Botrytis cinerea*, supporting a role in plant immunity[13]. Fungal LPMOs are another category of oxidoreductases that have been hypothesized to play a role in plant pathogenesis[46], although this is yet to be demonstrated despite the abundance of LPMO genes in most fungal phytopathogens.

Interestingly, the co-secretion of fungal LPMOs and AA7 enzymes has been observed[30,31], which inspired us to investigate the interplay between these two enzyme families. A balanced supply of ROS, e.g., $H_2O_2$ has been shown to be crucial for sustained LPMO activity, due to the affinity (low µM apparent $K_M$ value)[22] of LPMOs to $H_2O_2$ and their high sensitivity to oxidative damage[21,47]. We have shown that *Fg*CelDH7C potentiates LPMO-catalysed cellulose degradation in the absence of any exogenous reductant. The quenching of LPMO activity with either an alternative artificial acceptor for the AA7 (Supplementary Fig. 12) or an enzyme that scavenges $H_2O_2$ (Fig. 4a) is consistent with the proposed peroxygenase LPMO mechanism[21]. In addition, $O_2^{\bullet-}$ species that are generated by flavo-enzymes in the presence of $O_2$[48], may contribute to LPMO priming

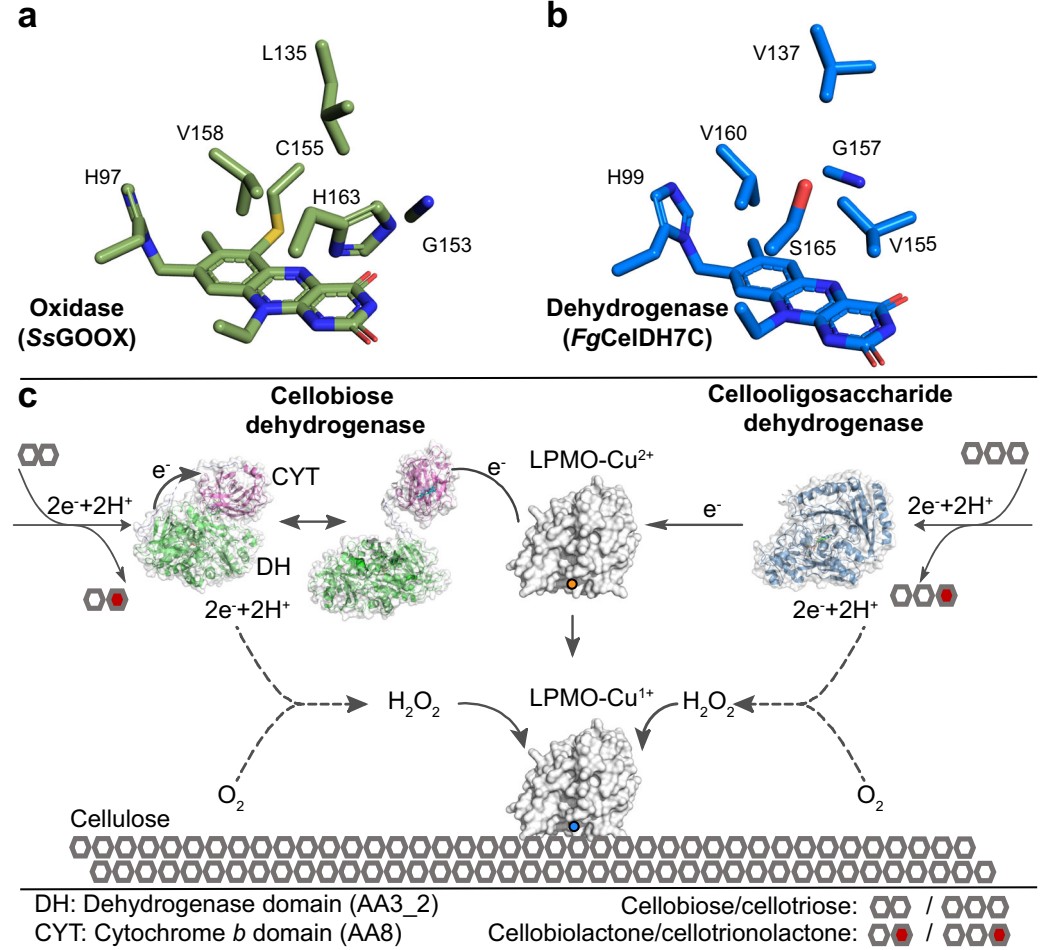

**Fig. 6 Oxygen-binding cavity in AA7 oxidoreductases and a schematic model for AA7-LPMO interplay. a, b** Putative oxygen-binding cavity in typical AA7 oxidases and dehydrogenases, respectively. The substitution of conserved histidine and glycine in oxidases to serine (or other small residues) and valine in dehydrogenases, respectively, is observed. **c** Schematic model of the interplay of cellobiose dehydrogenase (CDH) versus the AA7 cellooligosaccharide dehydrogenase with LPMOs during cellulose degradation. Both dehydrogenase classes oxidise cellooligosaccharides to the corresponding lactones. The electrons harvested from this oxidation are stored in the FAD-cofactor and subsequently delivered directly to the LPMO in the case of AA7. The transfer of priming electrons in CDH proceeds first from the dehydrogenase domain to the cytochrome $b$ haem domain in (closed form). A subsequent large conformational change to the open form is required to expose the haem domain to the LPMO active site for electron transfer. The dotted lines signify the low oxidase side-activity from both classes of dehydrogenases, which generates the $H_2O_2$ preferred co-substrate to fuel cellulose oxidative degradation by LPMOs. Low levels of $H_2O_2$ are also generated at the active site of free primed LPMOs, but this is left out from the figure for clarity.

consistent with previous findings[21,38]. To evaluate the specificity of the observed AA7-mediated activation of LPMOs, $Mo$Chi7A that possesses a low oxidase activity on cellooligosaccharides was used instead of $Fg$CelDH7C at equimolar concentrations in a similar assay. Under these conditions, the release of oligomers from cellulose was not observed to the same extent, indicating that the priming and/or the balanced fuelling of LPMO activity is more readily achieved with the AA7 dehydrogenase (Supplementary Fig. 15). Remarkably, the performance of $Fg$CelDH7C was comparable with CDH that has been shown to mediate direct transfer of priming electrons from cellooligosaccharide oxidation to LPMOs via the cytochrome $b$ domain[49]. Our EPR experiments were consistent with direct electron transfer to the LPMO when the FAD-cofactor in $Fg$CelDH7C was reduced either with dithionite or cellotriose in the absence of $O_2$ (Fig. 5c, d). This electron transfer and the interaction with the LPMO is likely to occur at the substrate-binding face ($si$-side) of the FAD, which is consistent with the rather open active site topology of $Fg$CelDH7C (Supplementary Fig. 9). By comparison, the electron

transfer from the larger (about 800 aa) and bi-modular CDH to the LPMO requires inter-domain electron transfer from the dehydrogenase to the cytochrome domain and a subsequent large domain movement to the open CDH form[49,50]. The deletion of the cytochrome domain impairs the priming of the LPMO, but the LPMO fuelling remains efficient[29], likely via the oxidase side-activity of CDH. Similarly, the deletion of the cytochrome domain in the pyrroloquinoline-quinone-dependent pyranose dehydrogenase impaired the fuelling of LPMO activity[24]. In summary, the one-module AA7 dehydrogenase confers direct priming of LPMOs in contrast to other known LPMO redox partners that require either small organic molecules or an additional cytochrome haem domain to mediate LPMO priming.

The expansion of the substrate range of fungal oligosaccharide-oxidising flavo-enzymes belonging to the AA7 family and the potentiation of LPMO activity, highlight the functional diversity within AA7. These findings set the stage for harnessing other AA7 dehydrogenases to fine-tune the activity of LPMOs that lack known redox partners. The demonstration of a functional and

efficient AA7-LPMO system offers attractive avenues for the development of commercial enzyme blends targeting recalcitrant biomass.

## Methods

**Bioinformatics and phylogenetic analysis**. The sequences of 2200 putative AA7 sequences were retrieved by a BlastP search against the non-redundant protein database interfaced the NCBI server (https://blast.ncbi.nlm.nih.gov/Blast.cgi) using the *Fg*ChitO sequence (Genbank accession: XP_011325372) as a query and default search settings. The retrieved sequences were filtered by excluding non-eukaryotic proteins and those displaying <25% amino acid (aa) identity and <90% coverage to the query. Two additional AA7-like sequences, which were differentially upregulated in the secretome of *Aspergillus nidulans* grown on starch[31], and a longer sequence possessing an N-terminal extension of about 250 amino acid residues (aa) (*Mo*Chi7A) were also included in the sequence inventory. Only sequences comprising 470–570 aa ($n = 1927$) were aligned using MAFFT[51] with default parameters, and curated using Gblock0.91b[52]. The alignment was used to construct a phylogenetic tree using NGphylogeny[53,54] and rendered by iTOL[54]. The visualization of amino acid conservation was made using WebLogo[55].

**Production of selected AA7 enzymes**. Based on the phylogenetic analysis above, five selected AA7 gene fragments encoding mature peptides without signal peptides (Supplementary Table 1) were codon optimised for *P. pastoris* and purchased from GENEWIZ (NJ, USA). The gene fragments were cloned within the XbaI and XhoI restriction sites of the pPICZαA vector (Invitrogen, Carlsbad, CA, USA) in frame with the *Saccharomyces cerevisiae* α-mating factor secretion signal and fused to a C-terminal (His)$_6$ tag. The synthetic gene-harbouring plasmids were propagated in *Escherichia coli* DH5α, linearised with PmeI and thereafter transformed into competent *P. pastoris* X33 by electroporation following the protocols from the Easy Select Expression System (Invitrogen). Six transformants per construct were screened for production in deep well plates for three days with MeOH addition 3% (v/v) every 24 h. The best-secreting transformants based on SDS-PAGE gel electrophoresis, from the constructs of the selected enzymes (Supplementary Table 1) were chosen for larger-scale production. These clones were grown in shake flasks containing 2 L of BMGY (Pichia expression manual, Invitrogen) containing 1 mL L$^{-1}$ of Pichia PTM4 trace element solution (2 g L$^{-1}$ CuSO$_4$.5H$_2$O, 3 g L$^{-1}$, MnSO4.H$_2$O, 0.2 g L$^{-1}$ Na2MoO$_4$.2H$_2$O, 0.02 g L$^{-1}$ H$_3$BO$_3$, 0.5 g L$^{-1}$ CaSO$_4$.2H$_2$O, 0.5 g L$^{-1}$ CoCl$_2$, 12.5 g L$^{-1}$ ZnSO$_4$.7H$_2$O, 22 g L$^{-1}$, FeSO$_4$.7H$_2$O, H$_2$SO$_4$ 1 mL L$^{-1}$) and biotin 0.2 g L$^{-1}$ to an $OD_{600}$ of 2–6, for 16 h at 30 °C using an orbital shaker (200 rpm). The cells from each construct were harvested by centrifugation (4000 × g, 10 min, 4 °C) and expression was induced by re-suspending the cells into 400 mL of BMMY medium at 20 °C and the culture was continued for 3 days with methanol supplementation to 3% (v/v) every 24 h. The cells were harvested by centrifugation (5,000 × g, 10 min, 4 °C) and the pH of the culture supernatants was adjusted to 7.8 using NaOH (2 M), followed by sterile filtration using 0.22 μm filters (Millipore, Burlington, MA, USA). The filtered supernatants were loaded onto 5 mL His Trap HP columns (GE Healthcare, Uppsala, Sweden) connected to an Äkta purifier 100 (GE Healthcare) at 5 mL min$^{-1}$, equilibrated with buffer A (Tris-HCl 50 mM pH 7.8, NaCl 150 mM, imidazole 10 mM). Non-bound proteins were washed by 10 column volumes (CVs). The bound proteins were eluted with a 34% buffer B (Tris-HCl 50 mM pH 7.8, NaCl 150 mM, imidazole 500 mM) gradient in 10 CVs. Fractions containing recombinant enzymes were pooled, concentrated and buffer exchanged against 50 mM NaOAc, pH 5.2 using a 10 kDa Vivaspin ultra-filtration unit (Sartorius, Göttingen, Germany). Protein purity was assessed using SDS-PAGE analysis and enzyme concentration was determined by measuring $A_{280}$ using a Nanodrop ND-2000 (Thermo Fisher Scientific, Waltham, MA, USA) and the theoretically calculated molar extinction coefficients (Table S1) using the Protparam tool on the EXPASY server (web.expasy.org/protparam/).

## Activity assays and kinetic analyses

*Initial screening assay*. The oxidase activity assay of the expressed *Mo*Chi7A, *Fg*Chi7B, *Fg*CelDH7C and *An*AA7A was conducted by coupling H$_2$O$_2$ production to HRP mediated oxidation of 4-aminoantipyrine (AAP) and 3,5-dichloro-2-hydroxybenzensulfonic acid (DCHBS) (both from Sigma Aldrich, St. Louis, MI, USA) in 96-well microtiter plates. Reactions (200 μL, 0.1 mM AAP, 1 mM DCHBS, 8 U mL$^{-1}$ HRP (Sigma Aldrich) in 50 mM NaOAc buffer, pH 5.2) were monitored by measurements of $A_{505\ nm}$ at 35 °C for 20 min with 30 s intervals. The assay was performed in triplicates using 0.1 μM AA7 against 30 carbohydrates, two sugar alcohols (all 2 mM except 1 mM for LNB/GNB) and 6 aromatic alcohols (1 mM) (Supplementary Table 2) by the addition of 100 μL substrate into pre-temperated reaction mixtures in microtiter plates. The substrate screening of *Pb*Chi7A was conducted using a modified reaction set up (250 μL, 0.1 mM AAP, 1 mM phenol, 8 U mL$^{-1}$ HRP (Sigma Aldrich) in 50 mM Tris-HCl pH 8.0) incubated in triplicates at 37 °C for 20 min and measured sequentially at $A_{505\ nm}$ with 30 s intervals.

*Oxidation kinetics of AA7*. The steady-state apparent kinetic parameters of *Mo*Chi7A and *Fg*Chi7B (0.65 μM) were determined in an analogous assay, where the reaction mixtures (200 μL) were monitored for 3 min with 20 s measurements intervals. The initial rates for the oxidation of N-acetylglucosamine (GlcNAc) and N-acetylgalactosamine (both in 0.09–12.5 mM) (GalNAc) (only for *Mo*Chi7A), chitobiose (0.0–0.5 mM) and chitotriose (0.03–0.6 mM) were determined in triplicates. The kinetic parameters of *Pb*Chi7A were determined using the same HRP assay on 12 substrates, including mono- (1–1500 mM), di- (1–250 mM), tri-saccharides (0.05–25 mM) (Supplementary Table 2). The kinetic parameters were determined by fitting the Michaelis-Menten equation in Origin, versions 9.55 and 18 (Northampton, Massachusetts, USA) to the initial rates, determined from the slopes of the linear reaction phases (Table 1). Due to the low oxidase activity of *Fg*CelDH7C, the activity assay was performed at higher enzyme concentrations (150 and 500 μM) in duplicates each and the absorbance was measured for one hour with 1 min intervals to determine the normalised reaction rate ($V_o$/E).

*Dehydrogenase assay of AA7 activity*. The dehydrogenase activity of the enzymes was also assayed using the redox mediator 2,6-dichlorophenolindophenol (DCIP) ($\varepsilon_{520} = 6.8$ mM$^{-1}$ cm$^{-1}$) as a terminal electron acceptor (as opposed to O$_2$ for the oxidase activity). Reactions (200 μL, 0.1 mM DCIP, 5 mM carbohydrate or 1 mM alcohol substrates, 0.65 μM enzyme in 50 mM HEPES, pH 7.0) were initiated by adding 100 μL carbohydrates or alcohol substrates to 100 μL pre-temperated mixtures containing all other components. The decrease in $A_{530}$ was measured continuously for 20 min with 30 s intervals at 35 °C.

*Temperature and pH stability/optimum measurements*. The pH-dependence of the activity of *Mo*Chi7A, *Fg*Chi7B and *Fg*CelDH7C (100 nM for all) towards 2 mM of GlcNAc, chitobiose and cellobiose, respectively, was analysed in triplicates using the dehydrogenase assay as above (DCIP as a terminal electron acceptor). The assay was carried out at eight pH values in the range 5.1–9.0 by exchanging the buffer in the screening assay to the universal Britton-Robinson buffer (acetic acid, phosphoric acid, and boric acid, 50 mM of each). The dehydrogenase assay was used to allow for comparison of all three enzymes in the same assay and the initial rates of the oxidation at each pH value we determined from the linear parts of the progress curves as above. Differential scanning calorimetry was employed to examine the conformational stability of *Fg*CelDH7C (1 mg mL$^{-1}$) at pH values of 5.2 and 7.2 in 50 mM (in acetate and phosphate buffers, respectively) in the temperature range of 15–90 °C at a scanning rate of 1 °C min$^{-1}$ using a NanoDSC instrument (TA instruments, New Castle, DE, USA).

The temperature and pH optima of *Pb*Chi7A (15 nM) were determined using the HRP assay described above against chitobiose (2 mM). For determination of the pH optimum, a Britton Robinson buffer (100 mM each) was adjusted to pH 4–11 and the oxidation activity was measured after 1 h incubation at 4 °C. The temperature stability of *Pb*Chi7A (15 nM) was measured by oxidation of 2 mM chitobiose within 1 h at different temperatures (25 °C–55 °C).

**NMR Spectroscopy of *Fg*CelDH7C, *Fg*Chi7B and *Mo*Chi7A**. Initially, the analysis of post oxidation reactions was conducted on a reaction mixture containing *Fg*Chi7B or *Mo*Chi7A (1 μM of each), 25 mM GalNAc or GlcNAc as substrates, buffered with 50 mM NaOAc buffer pH 5.2 (500 μL, 25 °C, 30 min). The reaction mixtures were ice-cooled and immediately analysed at 15 °C. Spectra were collected using an 800 MHz Bruker Avance II spectrometer equipped with an Oxford 18.7 T magnet and a TCI cryoprobe by transferring 500 μL of the post-reaction mixtures to NMR sample tubes (5 mm) and performing $^1$H, $^1$H-$^1$H TOCSY, $^1$H-$^1$H COSY, $^1$H-$^{13}$C HMBC NMR and multiplicity-edited $^1$H-$^{13}$C HSQC NMR spectroscopy analyses. $^1$H-$^1$H TOCSY spectra were acquired as data matrices of 1024 × 256 complex data points sampling 128 and 64 ms in the direct and indirect dimensions, respectively. $^1$H-$^1$H COSY spectra were acquired as data matrices of 2048 × 128 complex data points sampling 320 and 20 ms in the direct and indirect dimensions, respectively, while $^1$H-$^{13}$C HMBC NMR spectra were acquired as data matrices of 2048 × 256 complex data points sampling 212 and 5.8 ms in the direct and indirect dimensions, respectively. Multiplicity-edited $^1$H-$^{13}$C HSQC spectra were collected by sampling the NMR spectra for 106 and 15.9 ms in the direct ($^1$H) and indirect ($^{13}$C) dimensions, respectively. Once intermediate and product identification was established (Supplementary Fig. 3), a series of $^1$H-$^{13}$C HSQC spectra was acquired at 15 °C to follow the conversion of GlcNAc and GalNAc by *Mo*Chi7A (1 μM) in 50 mM NaOAc buffer pH 5.2 after 5, 15, 60 and 180 min experiment time. The $^1$H-$^{13}$C HSQC spectra were collected by sampling the NMR spectra for 159 and 8.3 ms in the direct ($^1$H) and indirect ($^{13}$C) dimensions, respectively, with experiments of 19 min duration to yield the time course shown in Supplementary Fig. 4 upon integration of the C2H2 NMR cross peaks.

For cellobiose, a time series of $^1$H-$^{13}$C HSQC spectra for the enzymatic conversion was acquired using *Fg*CelDH7C (0.56 μM) to yield assignments of cellobionolactone and of cellobionic acid signals as displayed in Supplementary Figure 5. Time-resolved $^1$H NMR spectra sampling 16384 complex data points of the FID for 1.27 s were subsequently used to track the reaction kinetics for cellobiose (2 mM) conversion by *Fg*CelDH7C (0.56 μM) at 25 °C in the presence or in the absence of 1.3 mM DCIP as an electron acceptor using the same equipment as above. The $^1$H singlet signal for DCIP as well as signals for H5 of cellobionolactone and H2 of cellobionic acid were integrated alongside the H2 and

H3 signals for the reducing end of cellobiose. Signal areas were normalized to the number of contributing hydrogen atoms. The same buffer, enzyme concentration and same NMR setup were subsequently used to track the dependence of cellotetraose oxidation (6 mM cellotetraose) by *Fg*CelDH7C in the presence of 0.025, 0.05, 0.1, 0.2, 0.5 or 1 mM DICP. All NMR spectra were acquired, processed and integrated in Topspin 3.5 pl6 with ample zero filling in all dimensions.

**Crystallization and structure Determination of *Fg*CelDH7C and *Fg*Chi7B**. Crystallization screens PACT++ (Jena Bioscience, Jena, Germany), JCSG+ and Morpheus screens (Molecular Dimensions, Sheffield, UK) and optimisation plates of *Fg*CelDH7C (9.0 mg mL$^{-1}$, deglycosylated with EndoH, New England Biolab, Ipswich, MA, USA) and *Fg*Chi7B (10.5 mg mL$^{-1}$), both in 20 mM NaOAc pH 5.5, were performed with a Crystal Gryphon liquid handling robot (Art Robbins Instruments, Sunnyvale, USA). Drops containing equal volumes of protein and reservoir solutions were mixed. Crystals of *Fg*CelDH7C appeared in the F7 condition (NaSCN, 20% (w/v) PEG 3350), which was optimised by varying PEG content (12.5–25% w/v). The best crystals formed within 4 days at 20 °C, using 150 nL of reservoir and protein solutions (1:1 ratio) in 50 µl reservoir solution. These crystals were cryo-protected with PEG400 in reservoir solution and cryo-cooled in a nylon loop. Crystals of *Fg*Chi7B were initially observed in PACT++ condition E10 (PEG 3350, 22% w/v, 200 mM sodium potassium phosphate using a 1:1 protein:reservoir ratio. Crystals were optimised by micro-seeding using the above condition at 22 and 24% PEG (w/v). Drops contained 1:1:0.2 of protein:reservoir: seed stock (100 fold diluted seed stock) ratio, respectively. The crystals were cryo-cooled using a mixture of reservoir and PEG400 (0.7:0.3 ratio) added to the drop prior freezing in liquid nitrogen. Diffraction data for *Fg*CelDH7C and *Fg*Chi7B crystals were collected in BioMax beamline at MaxIV (Lund, Sweden) using MXCuBE v3 and id30A3 (MASSIF-3) beamline at ESRF (Grenoble, France) using to 1.64 Å and 2.38 Å resolution, respectively. Both structures were solved using molecular replacement in Phaser[56] with model coordinates of *Tt*XylO] and initial automated model building and refinement using PHENIX.autobuild[57] and PHE-NIX.refine[58], respectively. Manual model rebuilding and map inspection were performed in Coot[59] and analysed using MolProbity[60]. The final validated structural models of *Fg*CelDH7C and *Fg*Chi7B were deposited to the Protein Data Bank under the entry codes 6YJI and 6YJO], respectively (Data collection and refinement statistics are in Supplementary Table 4).

**Cellulose degradation assays based on the interplay between LPMOs and AA7 enzymes**. The assays were performed using suspensions of either 1% PASC[61] and/or 0.5% Avicel (Honeywell Fluka, Morris Plains, NJ, U.S.A) (both w/v) in 250 or 500 µL containing 4.4 µM of *Pa*LPMO9E[20], *Pa*LPMO9H[20] or *Ls*AA9A[35] as well as 0.4 µM *Fg*CelDH7C (0.2 µM in case of *Ls*AA9A) in 50 mM NaOAc, pH 5.2 in the presence of cellotetraose (DP4, 0.8 mM), unless stated otherwise. For the comparison of *Fg*CelDH7C and *Mo*Chi7A, 0.2 µM of each was used. Controls were also performed in absence of individual reaction components or in the presence of 0.6 mM DICP. Moreover, the AA7-mediated LPMO activity was compared to reactions in the presence of ascorbate (1 mM), L-Cys (1 mM), reactions containing horseradish peroxidase (HRP, 0.2–38 nM in the presence of 0.1 mM AAP, 1 mM DCHBS), superoxide dismutase (SOD, recombinantly expressed in *E. coli*, 0.07–0.5 µM, Sigma Aldrich) in 100 µM phosphate, pH 7.0, or CDH from *Podospora anserina* (*Pa*CDH, 1.2 µM)[34].

The reactions were performed in 2 mL Eppendorf tubes with 850 rpm stirring in thermomixer (Eppendorf, Hamburg, Germany) at 35 °C for 2–24 h and thereafter quenched by boiling (100 °C, 10 min) and/or fourfold dilution in NaOH to 0.1 M followed by centrifugation (15,000 × g, 15 min, 4 °C) to separate the released saccharides in the supernatants from the insoluble fraction prior to analysis. The solubilised oxidised and non-oxidised oligosaccharides, released by the enzymatic treatments, were analysed by high-performance anion-exchange chromatography coupled with pulsed amperometric detection (HPAEC-PAD) (ThermoFischer Scientific) using a CarboPac™ PA1 (ThermoFischer Scientific) and a CarboPac™ PA1 guard column (2 × 50 mm) at 0.25 mL min$^{-1}$ and an eluent system as described by Westereng et al.[62]. The Chromeleon 6.80 SR10 software was used to interface the instrument and analyse the chromatograms. Non-oxidised oligosaccharides were used as standards (Megazyme, Wicklow, Ireland), whereas the corresponding C1-oxidised standards (from DP2-DP6) were provided from native cello-oligosaccharides using *Pa*CDHB treatment[34].

**EPR spectroscopy**. EPR experiments were performed on a Bruker ELEXSYS E500 X-band spectrometer equipped with an ER4102ST standard rectangular Bruker EPR cavity fitted to an Oxford Instruments ESR 900 helium flow cryostat. All EPR studies were carried out at 50 K with 4 mW microwave power at 9.479 GHz and 3 mT modulation amplitude at 100 kHz. The presented spectra were averaged from the accumulation of 4 scans to improve the signal-to-noise ratio. The EPR samples were prepared under anaerobic conditions in glove box (Jacomex) and frozen prior to the EPR analysis. In a first experiment, two 160 µL protein samples were prepared in 50 mM NaOAc pH 5.2 and incubated at 25 °C for 5 min: (i) control mixtures of *Pa*LPMO9H (100 µM) and cellotriose (DP3, 1 mM), (ii) same as (i), but supplemented with *Fg*CelDH7C (10 µM).

In the second set of experiments, *Pa*LPMO9H (20 µM) was prepared in the same buffer in the absence or presence of 40 µM dithionite and incubated at 25 °C for 15 min to achieve the full reduction of the catalytic Cu(II) center of *Pa*LPMO9H into Cu(I) by dithionite. Next, the following solutions were prepared and incubated anaerobically for 5 min at 25 °C: (i) *Fg*CelDH7C (20 µM) with sub-stoichiometric amount of sodium dithionite (16 µM) to reduce 80% of the flavo-enzyme, while avoiding excess dithionite in the solution, (ii) *Pa*LPMO9H (20 µM) without additives in the same buffer. Equal volumes of the two solutions were then pooled into a 160 µL EPR sample and incubated for few minutes prior freezing and EPR analysis.

**Reporting summary**. Further information on research design is available in the Nature Research Reporting Summary linked to this article.

## Data availability
The atomic coordinates of *Fg*CelDH7C and *Fg*Chi7B have been deposited in the Protein Data Bank (https://www.rcsb.org) under the PDB accessions 6YJI and 6YJO, respectively (see also Supplementary Table 4). The GenPept accession IDs of the enzymes characterised in the study are given in Supplementary Table 1. All the data are available from the corresponding authors upon request. Source data are provided with this paper.

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

## Acknowledgements

This study was supported by a Novo Nordisk Foundation Post-doc grant within the Biotechnology Based Synthesis and Production Research Programme to MHM (NNF17OC0025642). F.F. and D.H.W. were funded by The Novo Nordisk Foundation Grant No. NNF10CC1016517. NMR spectra were acquired at the NMR center DTU supported by the Villum and Carlsberg Foundations. J.G.B., B.G. (Bruno G) and F.B. (Frédéric B) thank IM2B (Institut de Microbiologie, Bioénergies et Biotechnologie). The DSC instrument is funded by the Carlsberg Foundation Grant (2013-01-0112) to M.A.H. The authors are also grateful to the EPR facilities available at the French EPR network (IR CNRS 3443) and the Aix-Marseille University EPR center. Prof. Peter Westh and Dr. Jeppi Kari are acknowledged for discussion on the redox-activities of AA7.

## Author contributions

M.H.M., J.G.B. and M.A.H. conceived the research and designed the experiments. M.H.M. and F.F. crystallised the enzymes, F.F. collected the crystallographic data and generated the first models, M.H.M. refined the structures with F.F. M.H.M. performed the sequence and phylogenetic analysis with help from V.L. Recombinant enzyme production was performed by M.H.M. and M.H. M.H.M. generated the initial biochemical characterisation except for *Pb*Chi7A. M.H.M. and T.S.N. carried out the kinetic analyses and TSN measured the pH, temperature and DCIP activity profiles. All biochemical data on *Pb*Chi7A were generated by O.R. and T.V.V. ERM provided the funding and the facilities for the work on *Pb*Chi7A. The NMR analyses were performed and interpreted by S.M. MHM performed the cellulose degradation assays and the HPAEC-PAD with support from S.G. M.H.M. and B.B. designed the SOD/HRP effect experiments, which were carried out by M.H.M. B.B. and M.H.M., F.B. and B.G. performed the EPR analysis. D.H.W. and B.H. provided funding and interpretation of the crystallography and sequence analysis part. M.A.H. and M.H.M. analysed the data and wrote the first version of the manuscript. J.G.B. and B.B. contributed to the writing of later versions of the manuscript. All authors contributed and approved the final version of the manuscript.

## Competing interests

A provisional patent has been filed based on data from this paper by M.H.M., J.G.B. and M.A.H. The other authors declare that they have no competing interests.
