## [Peer Review File · Nature Communications]

REVIEWER COMMENTS

Reviewer #1 (Remarks to the Author):

The manuscript by Momeni et al. is a well-executed investigation into a family of flavoenzymes (identified as AA7 in the CAZy database) that is particularly abundant in fungi, oomycetes and plants. The authors have integrated a comprehensive phylogenetic analysis with in vitro and structural characterisation of selected members, shedding new light on the structural basis underpinning different types of activity (oxidase vs dehydrogenase) seen within this enzyme family. The authors have characterised the activity of selected AA7 enzymes on a range of substrates, identified FgCelDH7C as an unusual dehydrogenase active on cello-oligosaccharides and shown its ability to deliver electrons to (previously characterised) AA9 LPMOs active on cellulose.

The range of experimental techniques and the quality of the data is commendable, and the authors make a number of interesting observations, however I feel that the story as a whole is not really ground breaking nor some of the findings entirely surprising. Several AA7 enzymes have been characterised both biochemically and structurally, and are known to catalyse the oxidation of several types of mono and oligosaccharides (including cello-oligosaccharides, xylo-oligosaccharides, chito-oligosaccharides, oligo-galacturonides, lactose) into their lactones, although the authors here have dug deeper into the some specific aspects. The other reason of the limited novelty of current work is that several classes of redox proteins (notably FAD-dependent AA3 cellobiose dehydrogenases and AA12 pyrroloquinoline-quinone-dependent oxidoreductases) have been previously shown to be good electron donors for LPMOs, and adding AA7 to the list does not feel to me particularly revolutionary (nor unexpected). FgCelDH7C seems to be better than fungal CDH at driving one specific LPMO tested in this study, but does not necessarily mean it will do that with any LPMO.

Hence I feel that this (nice) piece of work would be better suited for a more specialised journal.

Suggestions and specific comments:

- Line 56-58: not sure that this statement is correct. Unless I am mistaken, besides the four characterised fungal AA7s included in the CAZy database, several other non-fungal AA7s have been characterised before, although not included in CAZy (no idea why). For example AA7s from plants, see Benedetti et al, 2018, *The Plant Journal*, and Locci et al., 2019, *The Plant Journal*. This should be at least clarified in the text.

- Line 219-220: typo ("is that is that")

- Line 415: should be (Fig. 6, not Fig. 5).

- Supplementary Fig. 1: I have a few issues with this figure and the way the data are shown. The order in which the various substrates are listed in panel a vs c vs e (referring to the oxidases activities) is not consistent and should be corrected (for example, panel a shows cellobiose followed by maltose, while panel c inverts the order). Also, substrate GNB is in panel b and d but is missing from panel a and c. Also there are some substrates in panel e (e.g. ManNAc and galactosamine) that

are not in panels a-d, and viceversa some are absent from panel e (for example LNB and GNB). And then panel f (showing dehydrogenase activity) has yet a different range of substrates compared to e (which should be the corresponding oxidase activity). It would also be nice to show in panel e the dehydrogenase activity on cellobiose, which in panel e seems like a decent substrate for the oxidase activity of PbChi7A. In summary, there should be better consistency between the various panels of this figure.

Reviewer #2 (Remarks to the Author):

Referee report on "Discovery of fungal oligosaccharide-oxidising flavo-enzymes with previously unknown substrates, redox-activity profiles and interplay with LPMOs"

Momeni et al. has undertaken a detailed and thorough study of several fungal oligosaccharide-oxidizing flavo-enzymes capable of driving LPMO action. The study is well-planned and executed. There is no doubt that our understanding of LPMO action in nature and possibilities to fuel these enzymes in applied settings is greatly enhanced by this work. I strongly recommend publication in Nature Communications. Still, I would appreciate that two small issues were addressed as described below.

Minor comments:

1. Page 12.: Although the interplay between FgCelDH7C and PaLPMO9E and PaLPMO9H, respectively, is exquisitely demonstrated and described, I do struggle with the use of the term synergy. Especially how the results of the interplay or synergy is compared with the use of molecular oxygen in the presence of ascorbate for LPMO action. Synergy is an interaction or cooperation giving rise to a whole that is greater than the simple sum of its parts. Such synergy can be quantified using i.e. Abbot's equation for synergy calculation. FgCelDH7C fuels LPMO action by producing H₂O₂, which is undoubtedly a better co-substrate than O₂ with respect to enzyme speed (Bissaro et al. Nat. Chem. Biol 2017 and Kuusk et al. J. Biol. Chem. 2018). When O₂ is used as a co-substrate, the rate of LPMO action is either dependent on an innate monooxygenase reaction i.e. building up the reactive oxygen species at the active site during turnover with substrate present or the LPMOs oxidase activity (reducing O₂ to H₂O₂), which in turn is dependent on i.e. substrate concentration (equilibrium concentration of unbound LPMO), pH and so on. Without going into a further discussion or claiming which of these two scenarios describes the action of the AA9 in the presence of O₂ and ascorbate, still, to a fairly large extent, the authors are comparing apples and oranges. I am fine with the authors claiming synergy, but the above issues should be mentioned with respect to the discussion with respect to observed differences in enzyme speed.

2. Page 14: "The fact that only partial reduction of the Cu(II) center was observed could be due to kinetic effects." This may be correct, but there is also a possibility that there is a thermodynamic reason. AA9 tend to have redox potentials of 0.19 to 0.23 V. What the potential of FgCelDH7C is I do not know, but the pre-incubation experiment with celotriose and the pre-reduction with dithionite is in accordance with a close proximity of redox potentials. Vanillyl alcohol oxidase, as mentioned in the Introduction, has a reported redox potential of 0.055 V (van den Heuvel et al. J. Biol. Org, 2000), which is not too far from AA9 LPMOs.

Reviewer #3 (Remarks to the Author):

Deeper comprehension of structure and activity of AA7 family members is important for understanding of the strategies employed by microorganisms for efficient lignocellulosic biomass enzymatic degradation and can have potential biotechnological applications.

The manuscript describes production, biophysical, structural and biochemical characterization of a set of recombinant AA7 enzymes from different clades within AA7 family. Two crystal structures were solved and one enzyme was studied as an enzymatic tool to activate LPMOs. This is a nicely written and comprehensive manuscript, which can be further improved by following suggestions presented below:

1. Based on bioinformatics analysis, the authors subdivided AA7 family into 6 clades (I to VI) with some of them (clades II and V) being further subdivided into subclades A and B. They also state that all the characterized members of AA7 family map to the branch Va and lack of characterized members from clades II, III, IV, Vb and VI. Next the authors decided to focus on clades I, IIa, Va&b and VI. What is a rationale for such choice of the enzymes to study? If the idea was to make all the (uncharacterized) branches, some of them were not covered and some of the more characterized ones (Va) are present.

2. The authors determined crystal structures of FgChi7B (clade Va) and FgCelDH7C (clade IIa) and analyze in a considerable depth FgCelDH7C structure. Did the authors try to solve the structures of other expressed AA7 members? Why so little attention was given to FgChi7B? Can the structure be discussed in more detail?

3. FgCelDH7C was further chosen to fuel LPMO reaction. Did the authors try to activate LPMOs using other members of AA7 family? What was the result?

4. The authors show that both HPR and SOD, being added to LPMO+ FgCelDH7C+C3 reactions will cause dose-dependent decrease in LPMO activity (Fig. 5a). What is their interpretation of these results? The authors do not say if they added HRP substrate to the reaction (ABTS, AmplexRed, etc). How did the reaction proceed and what was the outcome?

5. SOD would act on dismutation of the superoxide radical into molecular oxygen and hydrogen peroxide. This should increase hydrogen peroxide concentration, which is a co-substrate of the LPMO. Nevertheless, the authors show that addition of SOD will lead to the decreased LPMO activity (Fig. 5b). One possible explanation of these results is that superoxide performs the LPMO priming reaction. However, the same suppression happens when DCIP is added to the reaction. The authors claim that the latter result indicates direct electron transfer from FgCelDH7C to LPMO and make a fair attempt to prove this using EPR. However, this apparently contradicts their own results with SOD. How the authors reconcile these observations?

Response to referees

We are very grateful to the constructive feedback and thorough work. Please find our detailed responses below:

Reviewer #1:

The manuscript by Momeni et al. is a well-executed investigation into a family of flavoenzymes (identified as AA7 in the CAZy database) that is particularly abundant in fungi, oomycetes and plants. The authors have integrated a comprehensive phylogenetic analysis with in vitro and structural characterisation of selected members, shedding new light on the structural basis underpinning different types of activity (oxidase vs dehydrogenase) seen within this enzyme family. The authors have characterised the activity of selected AA7 enzymes on a range of substrates, identified FgCelDH7C as an unusual dehydrogenase active on cello-oligosaccharides and shown its ability to deliver electrons to (previously characterised) AA9 LPMOs active on cellulose. The range of experimental techniques and the quality of the data is commendable, and the authors make a number of interesting observations, however I feel that the story as a whole is not really ground breaking nor some of the findings entirely surprising. Several AA7 enzymes have been characterised both biochemically and structurally, and are known to catalyse the oxidation of several types of mono and oligosaccharides (including cello-oligosaccharides, xylo-oligosaccharides, chito-oligosaccharides, oligo-galacturonides, lactose) into their lactones, although the authors here have dug deeper into the some specific aspects.

Response: We are very grateful for the thorough and constructive feedback. However, we respectfully do not share the same view regarding the novelty of the work based on: 1) All hitherto characterized AA7s are oxidases with a bi-covalently bound FAD, which has been the defining hallmark of this family. Our work changes the family definition, by revealing new clades and members with diverse modes of FAD-anchoring, substitutions in the FAD-domain "oxygen cage" and the correlation of these elements with a previously unknown oligosaccharide dehydrogenase activity. We are not sure if these findings can be considered as obvious without the comprehensive sequence, structural and biochemical work we present. 2) Our work identifies active site signatures that correlate with specificity towards carbohydrate/non-carbohydrate substrates and notably the identification of signatures for the different redox-profiles in this family, is another novel aspect of the work that considerably expands the inventory of AA7.

The other reason of the limited novelty of current work is that several classes of redox proteins (notably FAD-dependent AA3 cellobiose dehydrogenases and AA12 pyrroloquinoline-quinone-dependent oxidoreductases) have been previously shown to be good electron donors for LPMOs, and adding AA7 to the list does not feel to me particularly revolutionary (nor unexpected).

Response: Thanks for this comment. However, we would like to kindly argue that we do not share the same view on the novelty of our findings, based on: 1) Importantly, both the PQQ-dependent enzyme (AA8-AA12) and the CDH (AA8-AA3) are bi-modular enzymes, where the electron transfer has been shown to be dependent on a cytochrome b haem N-terminal domain as opposed to direct electron from the single module AA7 flavo-dehydrogenase in the present work. 2) Indirect electron transfer via small organic molecules has been reported, but again this is different from the direct electron transfer, which, to the best of our knowledge, has not been explicitly shown before.

In summary, our work is the first to demonstrate direct electron transfer to the LPMO as opposed to haem or small molecule mediated transfer in the cases referred to by the reviewer. We have clarified these aspects in the revised manuscript to better highlight the novelty of the findings.

FgCelDH7C seems to be better than fungal CDH at driving one specific LPMO tested in this study, but does not necessarily mean it will do that with any LPMO. Hence I feel that this (nice) piece of work would be better suited for a more specialised journal.

*Response: We would like to respectfully point out that the statement is not fully accurate. Our experiments have not been performed on a single LPMO, but on two different enzymes: the C1-oxidising PaLPMO9E and the C4-oxidising PaLPMO9H (see L264-280 in the submitted manuscript and Supplementary Fig. 11). However, the CDH comparison was performed only on the C4-oxidising enzyme, due to the technical reasons explained in the manuscript, i.e., being able to attribute the C4 oxidised species to the activity of the LPMO. Nonetheless, to address the reviewer's comment we have performed a new AA7/LPMO experiment with a third LPMO, the previously characterised C4-oxidising AA9 LPMO from *Lentinus similis* (LsAA9A). This new experiment yielded a similar outcome and was added to the revised manuscript (as Supplementary Fig. 13). In conclusion, our previous and new experiments demonstrate that the new AA7 fuels the activity of three different AA9 LPMOs from two different fungal spp., suggesting the interplay of FgCelDH with LPMOs is not restricted to a specific LPMO.*

Suggestions and specific comments:

- Line 56-58: not sure that this statement is correct. Unless I am mistaken, besides the four characterised fungal AA7s included in the CAZy database, several other non-fungal AA7s have been characterised before, although not included in CAZy (no idea why). For example AA7s from plants, see Benedetti et al, 2018, *The Plant Journal*, and Locci et al., 2019, *The Plant Journal*. This should be at least clarified in the text.

Response: We thank the reviewer for the good comments. The statements in the introduction referred to the AA7 assigned enzymes. We are aware of and have cited the work on the other enzymes, where we argue that these cases fulfil the criteria that we have identified to be considered as AA7s. To clarify, the following statement has been added to the introduction:

“In addition, oligosaccharide oxidases from plants have been reported^{13, 14}, but are currently not assigned into AA7.”

*Here we cited Locci et al. (*Plant J* (2019) 98(3):540-554), but this enzyme is too divergent from canonical enzymes (<25% aa identity), so it did not fulfil our inclusion criterion in the tree. Please note that we have also communicated to the curator of the CAZy database, who indicated that the AA7 inventory will be expanded in as soon as their time permits to include additional sequences based on our work.*

- Line 219-220: typo (“is that is that”)

- Line 415: should be (Fig. 6, not Fig. 5).

Response: We are grateful for the comment and the issues are corrected in the revised version.

- Supplementary Fig. 1: I have a few issues with this figure and the way the data are shown. The order in which the various substrates are listed in panel a vs c vs e (referring to the oxidases activities) is not

consistent and should be corrected (for example, panel a shows cellobiose followed by maltose, while panel c inverts the order). Also, substrate GNB is in panel b and d but is missing from panel a and c. Also there are some substrates in panel e (e.g. ManNAc and galactosamine) that are not in panels a-d, and vice versa some are absent from panel e (for example LNB and GNB). And then panel f (showing dehydrogenase activity) has yet a different range of substrates compared to e (which should be the corresponding oxidase activity). It would also be nice to show in panel e the dehydrogenase activity on cellobiose, which in panel e seems like a decent substrate for the oxidase activity of PbChi7A. In summary, there should be better consistency between the various panels of this figure.

Response: We thank the reviewer for the good observation. We have re-made the a-d panels to present the substrates consistently. The different substrates included in panels "e" and "f" compared to panels a-d, is due to the fact that the enzymes have different levels of activities for the different substrates, so for each enzymes, only the substrates that yield a signal higher than the noise are included. New experiments have been performed to add the missing dehydrogenase data in panel f. For GNB, the experiment has been done only in one assay based on availability of this pricy, but less preferred disaccharide, the relative activity level can be inferred from the dehydrogenase assay.

Reviewer

#2:

Referee report on "Discovery of fungal oligosaccharide-oxidising flavo-enzymes with previously unknown substrates, redox-activity profiles and interplay with LPMOs"

Momeni et al. has undertaken a detailed and thorough study of several fungal oligosaccharide-oxidizing flavo-enzymes capable of driving LPMO action. The study is well-planned and executed. There is no doubt that our understanding of LPMO action in nature and possibilities to fuel these enzymes in applied settings is greatly enhanced by this work. I strongly recommend publication in Nature Communications. Still, I would appreciate that two small issues were addressed as described below.

Minor

comments:

1. Page 12.: Although the interplay between FgCelDH7C and PaLPMO9E and PaLPMO9H, respectively, is exquisitely demonstrated and described, I do struggle with the use of the term synergy. Especially how the results of the interplay or synergy is compared with the use of molecular oxygen in the presence of ascorbate for LPMO action. Synergy is an interaction or cooperation giving rise to a whole that is greater than the simple sum of its parts. Such synergy can be quantified using i.e. Abbot's equation for synergy calculation. FgCelDH7C fuels LPMO action by producing H₂O₂, which is undoubtedly a better co-substrate than O₂ with respect to enzyme speed (Bissaro et al. Nat. Chem. Biol 2017 and Kuusk et al. J. Biol. Chem. 2018). When O₂ is used as a co-substrate, the rate of LPMO action is either dependent on an innate monooxygenase reaction i.e. building up the reactive oxygen species at the active site during turnover with substrate present or the LPMOs oxidase activity (reducing O₂ to H₂O₂), which in turn is dependent on i.e. substrate concentration (equilibrium concentration of unbound LPMO), pH and so on. Without going into a further discussion or claiming which of these two scenarios describes the action of the AA9 in the presence of O₂ and ascorbate, still, to a fairly large extent, the authors are comparing apples and oranges. I am fine with the authors claiming synergy, but the above issues should be mentioned with respect to the discussion with respect to observed differences in enzyme speed.

Response: We are very grateful for the positive and constructive feedback. We agree completely with the insightful reviewer and acknowledge that the ascorbate and AA7 reactions are not comparable and are not valid to claim synergy. We merely performed the ascorbate assay, due to the fact that ascorbate is the most common exogenous reductant in hitherto published LPMO assays. Our reasoning for using the term synergy for the AA7-reaction is that LPMO action (especially in a natural secretomes containing cellulases) generates cellooligosaccharide substrates for the AA7, and the action of the AA7 on cellooligosaccharides would generate the H₂O₂ co-substrate for the LPMOs. Indeed, we observe this type of “synergy” in the lack of added cellooligosaccharides, but the total level of the released cello-oligomers is enhanced markedly with addition of cellooligosaccharides (mimicking the presence of a cellulase). We have, however, revised the text and rephrased to omit the term “synergy” to avoid any misunderstandings based on the kind feedback.

2. Page 14: "The fact that only partial reduction of the Cu(II) center was observed could be due to kinetic effects." This may be correct, but there is also a possibility that there is a thermodynamic reason. AA9 tend to have redox potentials of 0.19 to 0.23 V. What the potential of FgCelDH7C is I do not know, but the pre-incubation experiment with cellotriose and the pre-reduction with dithionite is in accordance with a close proximity of redox potentials. Vanillyl alcohol oxidase, as mentioned in the Introduction, has a reported redox potential of 0.055 V (van den Heuvel et al. J. Biol. Org, 2000), which is not too far from AA9 LPMOs.

Response: We thank the reviewer and agree that the non-stoichiometric reduction could be theoretically due to either kinetic or thermodynamic barriers, so this is added in the revised manuscript. We previously determined the redox potentials of the PaAA9E and PaAA9H used in this study to +155 mV and +326 mV respectively (Garajova, et al (2016) Sci. Rep. 6, 28276). The redox potentials of bi-covalently FAD canonical oxidases are in the +130 mV range, whereas abolishing the cysteinyl bond reduces the redox potential to about +50-60 mV (Winkler, A et al, J. Biol. Chem. 282, 24437-24443 and Huang, C.-H., et al (2008) JBC 283, 30990-30996). We expect that that FgCelDHA to have a redox potential in the +50 mV range, which is likely to be sufficient to allow the LPMO-Cu(II) to harvest the electron from the FAD-center (compare to redox potentials of the haem domain of CHDs which is in the +90 mM region (at pH 6). This argumentation is also added to the revised manuscript to aid the reader.

Reviewer #3 (Remarks to the Author):

Deeper comprehension of structure and activity of AA7 family members is important for understanding of the strategies employed by microorganisms for efficient lignocellulosic biomass enzymatic degradation and can have potential biotechnological applications.

The manuscript describes production, biophysical, structural and biochemical characterization of a set of recombinant AA7 enzymes from different clades within AA7 family. Two crystal structures were solved and one enzyme was studied as an enzymatic tool to activate LPMOs. This is a nicely written and comprehensive manuscript, which can be further improved by following suggestions presented below:

1. Based on bioinformatics analysis, the authors subdivided AA7 family into 6 clades (I to VI) with some of them (clades II and V) being further subdivided into subclades A and B. They also state that all the characterized members of AA7 family map to the branch Va and lack of characterized members from

clades II, III, IV, Vb and VI. Next the authors decided to focus on clades I, IIa, Va&b and VI. What is a rationale for such choice of the enzymes to study? If the idea was to make all the (uncharacterized) branches, some of them were not covered and some of the more characterized ones (Va) are present.

Response: We thank the reviewer for the good comment and we agree it would have been more elegant to include clade III and IV enzymes. We had initially put more emphasis on clade I that represents a large part of the family and we selected four additional sequences from different branches of clade I, but these were unfortunately not expressed in the yeast host Pichia pastoris. Thus, the fact that we did not cover all clades in the family was due to the practical limitation of a two years post-doc.

2. The authors determined crystal structures of FgChi7B (clade Va) and FgCelDH7C (clade IIa) and analyze in a considerable depth FgCelDH7C structure. Did the authors try to solve the structures of other expressed AA7 members? Why so little attention was given to FgChi7B? Can the structure be discussed in more detail?

Response: We did attempt to crystallise MoChiA, which has an interesting activity profile and low K_m value, but we did not obtain good enough crystals. The focus was on FgCelDH7C due to the fact that it is very different in many respects from canonical enzymes in the family. The structure of the previously characterised chito-oligosaccharide active FgChiO was recently published (Savino, S et al (2020) FEBS Lett. 594, 2819-2828. To address the reviewer's comment, we have performed a structural comparison to this enzyme to our FgChi7B and to other counterparts that do not accept an N-acetyl group at the C2, e.g. cello- and xylo-oligosaccharide active enzymes. This analysis is included in the revised manuscript and visualised as a Supplementary Fig. 16 to highlight the structural elements underpinning the recognition of different oligosaccharide substrates in AA7.

3. FgCelDH7C was further chosen to fuel LPMO reaction. Did the authors try to activate LPMOs using other members of AA7 family? What was the result?

Response: Indeed, MoChi7A (clade Vb), which has a side activity on cellooligosaccharides, displayed some extent of interplay with PaLPMO9H, but at a much lower level than that observed with FgCelDH7C. This was shown in the original submission as Supplementary Figure 14b (Supplementary Figure 15b in revised version).

4. The authors show that both HPR and SOD, being added to LPMO+ FgCelDH7C+C3 reactions will cause dose-dependent decrease in LPMO activity (Fig. 5a). What is their interpretation of these results? The authors do not say if they added HRP substrate to the reaction (ABTS, AmplexRed, etc). How did the reaction proceed and what was the outcome? SOD would act on dismutation of the superoxide radical into molecular oxygen and hydrogen peroxide. This should increase hydrogen peroxide concentration, which is a co-substrate of the LPMO. Nevertheless, the authors show that addition of SOD will lead to the decreased LPMO activity (Fig. 5b). One possible explanation of these results is that superoxide performs the LPMO priming reaction. However, the same suppression happens when DCIP is added to the reaction. The authors claim that the latter result indicates direct electron transfer from FgCelDH7C to LPMO and make a fair attempt to prove this using EPR. However, this apparently contradicts their own results with SOD. How the authors reconcile these observations?

Response: We thank the reviewer for the comment. The HRP assay included 0.1 mM AAP, 1 mM DCHBS

similar to the well-established activity assay that consumes H₂O₂ to create a DCHBS radical and oxidise APP.

Regarding the interpretation of the data, the depletion of H₂O₂ results in steep decrease and eventually loss of activity at nM concentrations of HRP. This is consistent with the preference for H₂O₂ as a co-substrate as compared to oxygen and the ability of AA7 to prime the LPMO and fuel the reaction via the supply of low levels of H₂O₂. With regard to the SOD reaction, dismutation results in the production of H₂O₂ equivalents. Therefore, the loss of activity is likely attributed to a specific role of the superoxide ion, which cannot be compensated by the potential increase in rate caused by higher H₂O₂ supply. It has indeed previously been shown that superoxide can carry out LPMO priming (Bissaro et al. (2017) Nat. Chem Biol; Bissaro et al. (2020) Nat. Comm.). Such role in our study thus cannot be excluded. On the other hand, EPR experiments, carried out under anaerobic conditions and showing partial reduction of the LPMO, are also indicative of the possibility of O₂-independent, direct reduction. As a conclusion, we agree with the reviewer that both hypotheses should be put on the table. Of note, the two different mechanisms are not contradictory, as either can contribute to LPMO priming and the partition between both ways under aerobic conditions will be mainly driven by thermodynamics.

We have amended the main text accordingly:

Old text:

In the presence of increasing concentrations (0.07-0.5 μM) of SOD³⁷, we observed a similar inhibitory trend. These findings suggest that H₂O₂ and O₂•- play a role in the FgCelDH7C-fuelled LPMO reaction (Fig. 5b). When assessing AA7-LPMO synergy, we noted that the addition of DCIP to the reaction mixture was inhibitory (Supplementary Fig. 11). We hypothesised that this result is indicative of electron transfer occurring from the AA7 to the LPMO, which is inhibited by the competitive electron acceptor DCIP. To provide evidence for such priming electron transfer, we used Electron Paramagnetic Resonance (EPR) spectroscopy. The extent of LPMO-Cu(II) reduction to Cu(I) was monitored as a decrease of the Cu(II) EPR signal due to silence of LPMO-Cu(I) species in EPR²¹.

New text:

In the presence of increasing concentrations (0.07-0.5 μM) of SOD³⁷, we observed a similar inhibitory trend (Fig. 5b). These findings suggest that O₂•- plays a specific role in the FgCelDH7C-fuelled LPMO reaction (Fig. 5b). Indeed, the priming reduction of LPMOs with the superoxide ion has been previously proposed³⁸. In addition to O₂•- mediated priming, we also investigated the possibility of direct priming electron transfer between the AA7 and the LPMO using Electron Paramagnetic Resonance (EPR) spectroscopy. The extent of LPMO-Cu(II) reduction to Cu(I) was monitored as a decrease of the Cu(II) EPR signal due to silence of LPMO-Cu(I) species in EPR²¹.

We thank all reviewers for their kind work.

REVIEWERS' COMMENTS

Reviewer #3 (Remarks to the Author):

The authors responded to all the questions of the referees in a satisfactory manner. I have no further comments or suggestions.